# NeoPrecis: enhancing immunotherapy response prediction through integration of qualified immunogenicity and clonality-aware neoantigen landscapes

Ko-Han Lee ®[1], Timothy J. Sears ®[1], Maurizio Zanetti[2,3] & Hannah Carter ®[1,3,4] ✉

Despite the transformative impact of cancer immunotherapy, the need for improved patient stratification remains critical due to suboptimal response rates. While neoantigens are central to anti-tumor immunity, current metrics, such as tumor mutation burden (TMB), are limited by their neglect of immunogenicity and tumor heterogeneity. Here we present NeoPrecis, a computational framework designed to improve immunotherapy response prediction by refining neoantigen characterization across MHC-I and MHC-II pathways and by integrating tumor clonality information. NeoPrecis features an interpretable T-cell-recognition model that reveals the critical influence of MHC molecules on TCR recognition beyond mere antigen presentation. Benefit HLA alleles, identified through model-driven contribution analysis, exhibit significant predictive power for patient outcomes in immune checkpoint inhibitor treatment (melanoma: $p$-value = 0.04; NSCLC: $p$-value = 0.01). NeoPrecis, via its clonality-aware neoantigen landscape feature, improves immunotherapy response prediction in tumor types with varying prevalence of neoantigens, including heterogeneous NSCLC, which retains more subclonal neoantigens due to lower immunoediting pressure. We thus propose NeoPrecis as a comprehensive evaluative framework for neoantigen assessment by incorporating both immunogenicity and tumor clonality, offering insights into the link between the collective quality of neoantigen landscapes and immunotherapy response.

Immunotherapy has revolutionized cancer treatment, with personalized cancer vaccines and immune checkpoint inhibitors (ICI) emerging as two major modalities targeting tumor-specific neoantigens[1,2]. Cancer vaccines work by boosting the immune system through the expansion of T cell precursors specific for these tumor-specific antigens. Since 2017, numerous cancer vaccine trials have been conducted in cancers such as melanoma[3–6], glioblastoma[7,8], pancreatic cancer[9],

and others[6,10,11]. However, response rates remain suboptimal. For instance, Rojas et al. reported an overall response rate of 50%, yet only 11% of target neoantigens successfully induced a T cell response[9]. These limitations underscore the need for improved strategies to identify and target immunogenic neoantigens more effectively.

Meanwhile, ICIs have demonstrated significant efficacy in certain cancer types, such as melanoma[12,13] and non-small cell lung cancer

[1]Bioinformatics and Systems Biology Program, University of California San Diego, La Jolla, CA, USA. [2]The Laboratory of Immunology, Department of Medicine, University of California San Diego, La Jolla, CA, USA. [3]Moores Cancer Center, University of California San Diego, La Jolla, CA, USA. [4]Department of Medicine, Division of Genomics and Precision Medicine, University of California San Diego, La Jolla, CA, USA. ✉e-mail: hkcarter@health.ucsd.edu

(NSCLC)[14,15], by blocking immune checkpoint pathways and reinvigorating CD8 + T cells recognizing tumor-specific antigens, including neoantigens. Despite their transformative potential, ICIs exhibit variable and often suboptimal response rates[16,17]. For instance, in melanoma, one of the most responsive cancer types, single-agent ICI achieves a response rate of approximately 40%, while combination therapy increases this to around 60%[18,19]. However, some patients experience life-threatening immune-related adverse events after receiving ICIs[20]. Tumor mutation burden (TMB), an FDA-approved biomarker for immunotherapy, acts as a proxy for immunogenic neoantigen abundance[21]. It is known to correlate with response rates across various cancer types[16]. High-TMB cancers like melanoma show higher response rates, while low-TMB cancers such as sarcoma have lower efficacy. Nevertheless, within individual cancer types, TMB's predictive power is limited[17]. Tumor neoantigen burden (TNB), which relates to the count of MHC-presented neoantigens, offers a more refined metric of the tumor's immune landscape but fails to account for T-cell receptor (TCR) recognition or the clonal distribution of mutations within tumors[22].

Neoantigens are novel peptide antigens derived from tumor-specific mutations, playing a critical role in the immune system's ability to recognize and target tumors. The immunogenicity of neoantigens is highly associated with their abundance and peptide binding affinity for the major histocompatibility complex (MHC), which displays them on the cell surface[23]. However, these factors alone are insufficient for a sensitive and efficient selection of neoantigens[24]. The activation of T cells requires TCR recognition of the peptide-MHC (pMHC) complex. Several indirect metrics have been proposed to evaluate TCR recognition, including the agretopicity ratio (the ratio of MHC-binding affinity between mutated and wild-type peptides)[23,25] whereby a mutant peptide could be more easily recognized by the TCR repertoire if the wild-type counterpart was not involved in thymic selection, and foreignness (measuring the sequence similarity to foreign immunogenic antigens)[26]. Direct approaches include ICERFIRE[27], DeepNeo[28], and PRIME[29,30], which predict immunogenicity based on peptide sequence, and cross-reactivity distance[31], which measures the distance between wild-type and mutated peptides in terms of TCR recognition. Despite these efforts, the efficacy of TCR recognition metrics remains limited due to the high diversity of TCR repertoires, shaped by genetic recombination and thymic selection[32,33]. Additionally, MHC allele variability across individuals leads to distinct peptide repertoires, further constraining the generalizability of predictive metrics, underscoring the need for improved methods.

Tumor clonality further shapes the immune response by influencing the distribution of neoantigens across tumor subclones. Clonal neoantigens, which are derived from mutations shared by all tumor cells, can generate more effective clinical responses than subclonal neoantigens, which are confined to a subset of tumor cells[34,35]. Highly heterogeneous tumors, marked by a greater proportion of subclonal mutations, may evade immune surveillance due to reduced clonal neoantigen exposure[36–38]. Thus, CD8 + T cells predominantly react to clonal mutations in early-stage NSCLC[34], and NSCLC tumors with a high proportion of subclonal copy-number alterations are at higher risk for recurrence or death than those with a low proportion[39]. Likewise, high tumor heterogeneity correlates with limited ICI response in both mice and humans in melanoma[40] and with poor survival outcomes in a meta-analysis of the TCGA cohort[38]. These findings suggest that suboptimal clinical outcomes may reflect the inability of the immune cells to cope with tumor heterogeneity and underscore the importance of identifying immunogenic neoantigens, as they are not only important biomarkers of ICI response but also indicators of tumor-immune co-evolution through immune selection.

Historically, neoantigen discovery has predominantly focused on the MHC-I pathway[41], which activates CD8 + T cells for direct tumor cell killing. This focus reflects the established role of CD8 + T cells as primary cytotoxic effectors in anti-tumor immunity. However, recent studies have expanded our understanding of CD4 + T cells, highlighting their critical roles in both direct tumor killing and CD8 + T cell priming. Bawden et al. revealed direct cytotoxic effects of CD4 + T cells against melanoma[42], while Espinosa-Carrasco et al. emphasized their role in sustaining CD8 + T cell responses through interactions with intra-tumor dendritic cells (DC)[43]. Further supporting this, Pyke et al.[44] and Alspach et al.[45] demonstrated the contribution of MHC-II neoantigens to anti-tumor immunity. Moreover, Sears et al.[46] found that patients with MHC-II-reliant neoantigen profiles, featuring a higher proportion of MHC-II neoantigens, experience significantly longer-lasting clinical benefits. These findings reinforce the critical role of MHC-II neoantigens and highlight the necessity of identifying both MHC-I and MHC-II neoantigens to fully capture the cooperative interplay between CD8+ and CD4 + T cells in tumor immunity.

To encompass these variables, we present NeoPrecis, a computational framework that predicts neoantigen immunogenicity and constructs a clonality-aware neoantigen landscape model to more accurately evaluate patients' potential immune responses (Fig. 1). NeoPrecis integrates three critical dimensions—neoantigen abundance, MHC presentation, and TCR recognition—into a unified evaluative framework that incorporates both MHC-I and MHC-II pathways. By accounting for MHC allele variability and tumor clonality, NeoPrecis provides a more precise representation of the tumor's immune landscape that addresses the limitations of current bulk metrics. This comprehensive approach improves the identification of immunogenic neoantigens and enables better patient stratification for immunotherapy.

## Results
### Characterizing T-cell recognition in immunogenicity prediction
Measures for assessing neoantigen abundance and binding to MHC molecules are well established. In contrast, predicting T-cell recognition is challenging due to the high variability of TCR repertoires and the still-limited availability of TCR sequencing data. To incorporate a model of T cell recognition into our in silico neoantigen prioritization pipeline, we considered two alternative approaches for assessing TCR recognition without TCR sequencing: peptide-based methods and the cross-reactivity distance (CRD) approach. Cross-reactivity, a well-documented feature of TCR-peptide binding, describes the ability of a TCR to bind multiple peptides with similar sequences[47]. Peptide-based methods, such as PRIME[29,30] and DeepNeo[28], primarily model the immunogenicity of target peptides. In contrast, CRD, first introduced by Łuksza et al.[31], measures the differences between wild-type (WT) and mutated (MT) peptides, aligning more closely with immune mechanisms, as thymic selection acts to constrain TCR diversity to distinguish self from non-self peptides[33]. However, existing models overlook the role of MHC-binding motifs, which are critical in shaping TCR-pMHC interactions. To address this limitation, we developed a new immunogenicity model for NeoPrecis, NeoPrecis-Immuno, which incorporates MHC-binding motifs into a refined CRD-based framework, enabling a more nuanced characterization of neoantigens.

NeoPrecis-Immuno embeds MHC-binding core regions (9-mer) using trained amino acid embeddings and MHC-binding motif enrichment, projecting each residue into a latent space specific to MHC alleles. Residue-level embeddings are then aggregated using position-weighted pooling to generate peptide embeddings for WT and MT peptides, highlighting the varying importance of core positions in TCR-pMHC interactions. To capture differences between WT and MT peptides, NeoPrecis-Immuno employs a geometric representation that includes origin, direction, and distance. The distance component predominantly reflects CRD and is further refined with peptide sequence information (origin and direction). Finally, NeoPrecis-Immuno integrates CRD with MHC-binding score as a covariate, allowing the model to further learn immunogenicity beyond

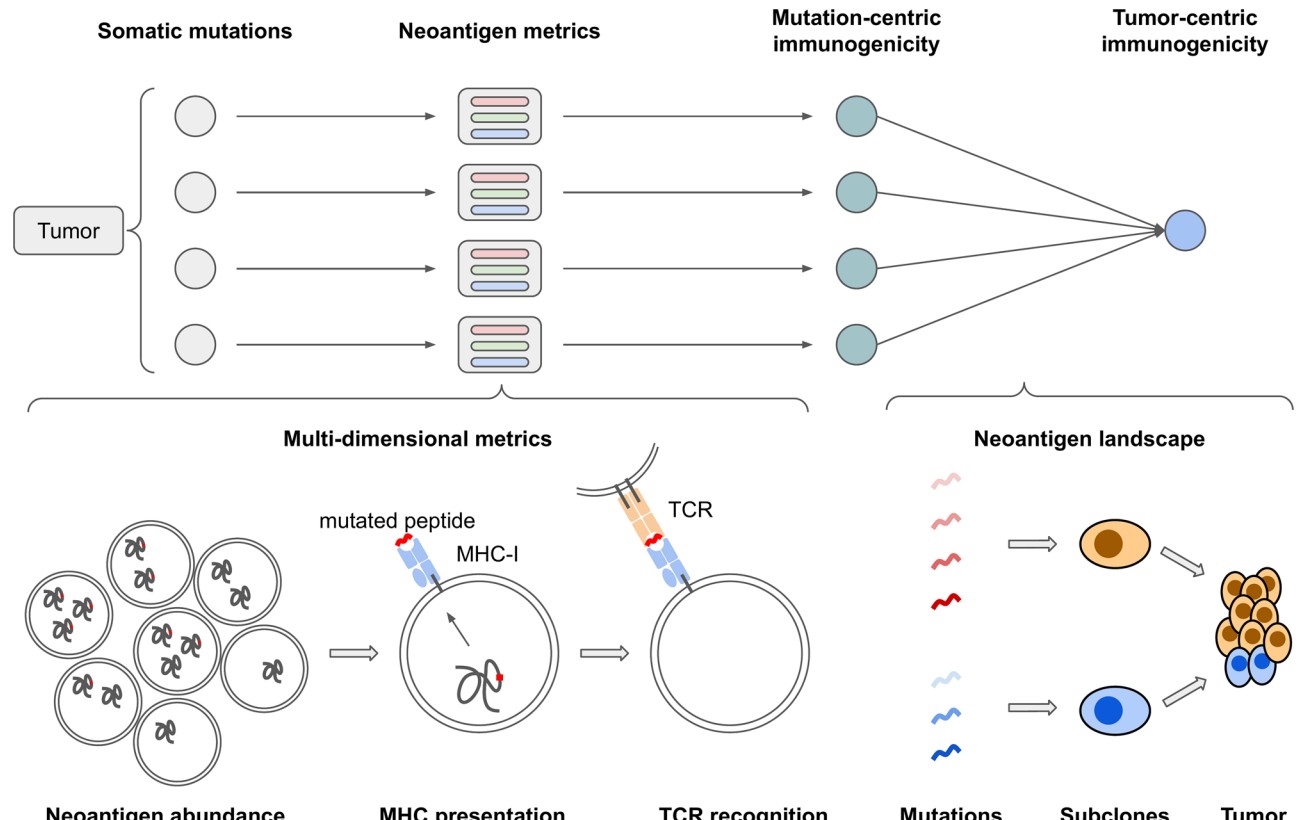

**Fig. 1 | Overview of NeoPrecis for tumor neoantigen assessment.** The workflow begins with the computation of neoantigen-related multi-dimensional metrics, including neoantigen abundance, MHC presentation, and TCR recognition, for each protein-altering somatic mutation. These metrics are subsequently integrated to derive mutation-centric immunogenicity scores. Finally, a tumor-centric immunogenicity score is generated using a framework designed to map the neoantigen landscape considering neoantigen immunogenicity and tumor clonality.

what is captured by MHC binding alone, to output a probability that represents neoantigen immunogenicity (Fig. 2a, Methods).

We implemented a two-stage training process using TCR-pMHC binding data from IEDB[48] and VDJdb[49] (Supplementary Data 1) and T-cell assay data from CEDAR[50] (Supplementary Data 2). In the first stage, we constructed a cross-reactive peptide triplet dataset from binding data, where each triplet consists of a seed peptide, a cross-reactive peptide (peptide sharing TCR binding with the seed), and a non-cross-reactive peptide (peptide not sharing TCR binding). We curated 1153 MHC-I and 261 MHC-II cross-reactive pairs and sampled 10 non-cross-reactive peptides for each pair, yielding 11,530 MHC-I and 2610 MHC-II triplets for training. The model was trained to minimize the distance between cross-reactive pairs (seed and cross-reactive peptides) and maximize the distance between non-cross-reactive pairs (seed and non-cross-reactive peptides) (Supplementary Fig. 1a, Methods). In the context of a tumor, mutant peptides more similar to wild-type (cross-reactive with the same TCRs) are less likely to be immunogenic because T cells recognizing these self-similar epitopes are typically eliminated during central tolerance. In contrast, more dissimilar mutants (less cross-reactive) are more likely to be immunogenic, as T cells capable of recognizing these neoantigens escape thymic negative selection and remain available in the peripheral repertoire. Notably, the geometric distance calculated by NeoPrecis-Immuno outperformed the BLOSUM62 and PMBEC (an amino acid similarity matrix for peptide-MHC binding)[51] distances in differentiating cross-reactive and non-cross-reactive peptide pairs (Fig. 2b, Supplementary Fig. 2a).

In the second stage, we fine-tuned the model using T-cell assay data with T-cell activation labels, enabling it to learn the immunogenicity of target peptides (Methods). We curated 4176 MHC-I

and 87 MHC-II samples predicted to bind MHC (NetMHCpan %rank ≤ 2 for MHC-I and ≤ 10 for MHC-II) from CEDAR and split them into training (75%), validation (10%), and testing (15%) sets. The resulting immunogenicity model was subsequently validated on both an internal and an external testing set to benchmark its performance.

We compared NeoPrecis-Immuno to an MHC-binding predictor (NetMHCpan[52,53]) and immunogenicity models (PRIME[29,30], ICERFIRE[27], and DeepNeo[28]) (Methods). The internal testing set, isolated from the CEDAR dataset[50] before training, is epitope-based with labels for each pMHC, while the external testing set, derived from an independent cancer cohort (NCI dataset, Parkhurst et al.[54]), is mutation-based with T-cell assay data for each mutation (Supplementary Data 3). For mutation-based predictions, aggregation across MHC alleles was required; we systemically evaluated aggregation strategies later in the Results (Methods). As there weren't many MHC-II datapoints, internal validation was performed exclusively for MHC-I. To ensure fairness, we identified pMHCs not present in the training datasets of any predictor (n = 1147, Supplementary Fig. 2b) and excluded pMHCs with unavailable MHC alleles in any predictor. Among the remaining 438 pMHCs (#positives = 228), NeoPrecis-Immuno outperformed NetMHCpan and DeepNeo but performed slightly worse than PRIME and ICERFIRE (Fig. 2c, Supplementary Fig. 2c). We also rebalanced samples through bootstrapping to achieve a 1:3 positive-to-negative ratio, matching the test set balance used in PRIME and ICERFIRE benchmarks and better reflecting realistic immunogenicity rates[55], which yielded similar performance rankings (Supplementary Fig. 2d). For external validation, NeoPrecis-Immuno achieved the best performance for both MHC-I (n = 1089; #positives = 36) and MHC-II (n = 1189; #positives = 33) predictions. For MHC-I, NeoPrecis-Immuno outperformed all other

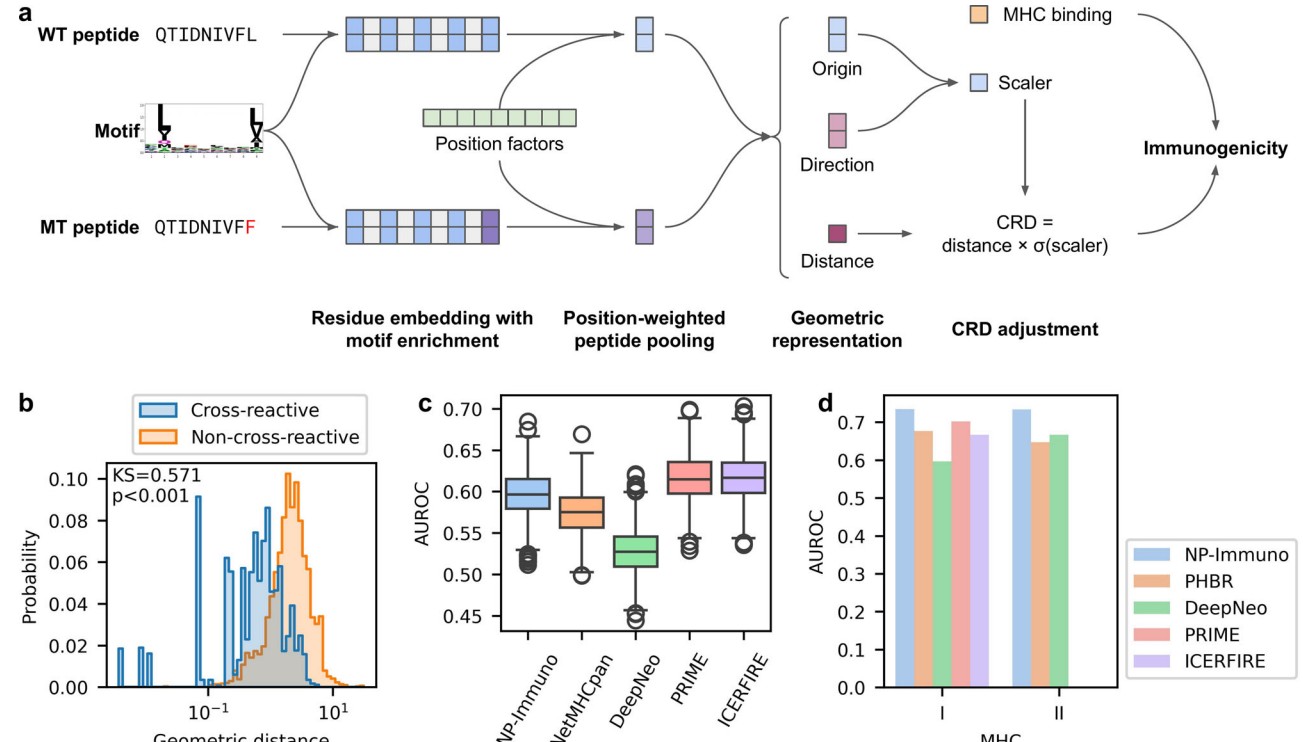

**Fig. 2 | Development and validation of the T-cell recognition model, NeoPrecis (NP)-Immuno. a** Schematic of NeoPrecis-Immuno architecture. Peptide residues are embedded with motif enrichment features, and their contributions are weighted by trainable position factors to generate a peptide embedding. A geometric representation captures the differences between wild-type (WT) and mutated (MT) peptides, from which the cross-reactivity distance is computed. Combined with the MHC binding prediction as a covariate, the model outputs an immunogenicity score highlighting T-cell recognition potential. **b** Distribution of geometric distances in the cross-reactive peptide triplet validation set. The cross-reactive group ($n = 3535$) consists of pairs of seed and cross-reactive peptides that mimic non-immunogenic substitutions, while the non-cross-reactive group ($n = 3535$) consists of pairs of seed and non-cross-reactive peptides that mimic immunogenic substitutions. Statistical comparison was performed using a one-sided Kolmogorov-Smirnov test (KS statistic = 0.571, $P < 1 \times 10^{-300}$). **c** Area under the receiver operating characteristic curve (AUROC) for each predictor on MHC-I samples ($n = 438$ pMHCs; #positives = 228) from the internal testing set (CEDAR dataset). AUROC distributions were calculated using bootstrapping (1000 iterations). In the boxplots, the center line represents the median, boxes indicate the interquartile range (IQR; 25th–75th percentile), and whiskers extend to the most extreme data points within 1.5× the IQR. Outliers are depicted as individual circles. **d** AUROC comparisons of each predictor on MHC-I ($n = 1089$ peptides; #positives = 36) and MHC-II ($n = 1189$ peptides; #positives = 33) samples from the external testing set (NCI dataset). PHBR is an allele-aggregating score for MHC-binding prediction.

predictors across AUROC, AUPRC, and PPV metrics. For MHC-II, NeoPrecis-Immuno achieved the highest AUROC while performing comparably to DeepNeo in AUPRC and PPV (Fig. 2d, Supplementary Fig. 2e). Although mutations with no available alleles were excluded, for some mutations, not all alleles were supported (<6 alleles for MHC-I; <10 alleles for MHC-II) by ICERFIRE and DeepNeo (Supplementary Fig. 2f), potentially contributing to their lower performance. Overall, these benchmarks demonstrate that NeoPrecis-Immuno is competitive with other immunogenicity predictors.

To assess model generalizability, we evaluated TCR diversity in the training data and performance on unobserved MHC alleles. For TCR diversity, we analyzed CDR3 coverage in the cross-reactive peptide triplet dataset and found that approximately 50% of cross-reactive peptide pairs are recognized by multiple distinct CDR3 sequences, with more than 50% of amino acid substitutions supported by multiple independent CDR3s (Supplementary Fig. 3a). This multi-CDR3 support indicates that the training data captures generalizable cross-reactivity patterns rather than isolated clone-specific interactions. For MHC allele generalization, we stratified mutations in the NCI dataset by the number of unobserved alleles—those absent from training data. The CEDAR test set contained only 2 unobserved alleles each for MHC-I and MHC-II (Supplementary Fig. 3b), necessitating the use of NCI for this evaluation. In NCI, 28% of mutations had unobserved MHC-I alleles, while nearly all mutations had at least one unobserved MHC-II allele (Supplementary Fig. 3c). Although performance decreased in the

unobserved allele group, NeoPrecis-Immuno outperformed other predictors for MHC-I and performed comparably for MHC-II (Supplementary Fig. 3d). We note that alleles unobserved in our training data may have been observed in other predictors' training sets, potentially contributing to their performance in these cases. These results demonstrate that NeoPrecis-Immuno generalizes effectively across unobserved MHC alleles and that the training data represent diverse TCR recognition patterns, though limited training diversity remains a constraint.

## Interpretable MHC-binding motif contributions to T-cell recognition

To evaluate the contribution of each component in NeoPrecis-Immuno, we extracted scores from the model for substitution distance (SubDist), substitution distance with position weighting (SubPosDist), geometric distance (GeoDist; SubPosDist with motif enrichment), and cross-reactivity distance (CRD; GeoDist with sigmoid scaling) (Supplementary Table 1, Methods). BLOSUM62 (BLOSUMDist) and PMBEC (PMBECDist) distances were included as baselines for comparison. These distances quantify the difference between WT and MT peptides. All NeoPrecis-Immuno components outperformed the baseline in both MHC-I and MHC-II contexts (Fig. 3a). For MHC-I, performance steadily improved as additional components were added to the model. However, in MHC-II, differences between components were less pronounced, likely due to the smaller sample size. Sigmoid scaling

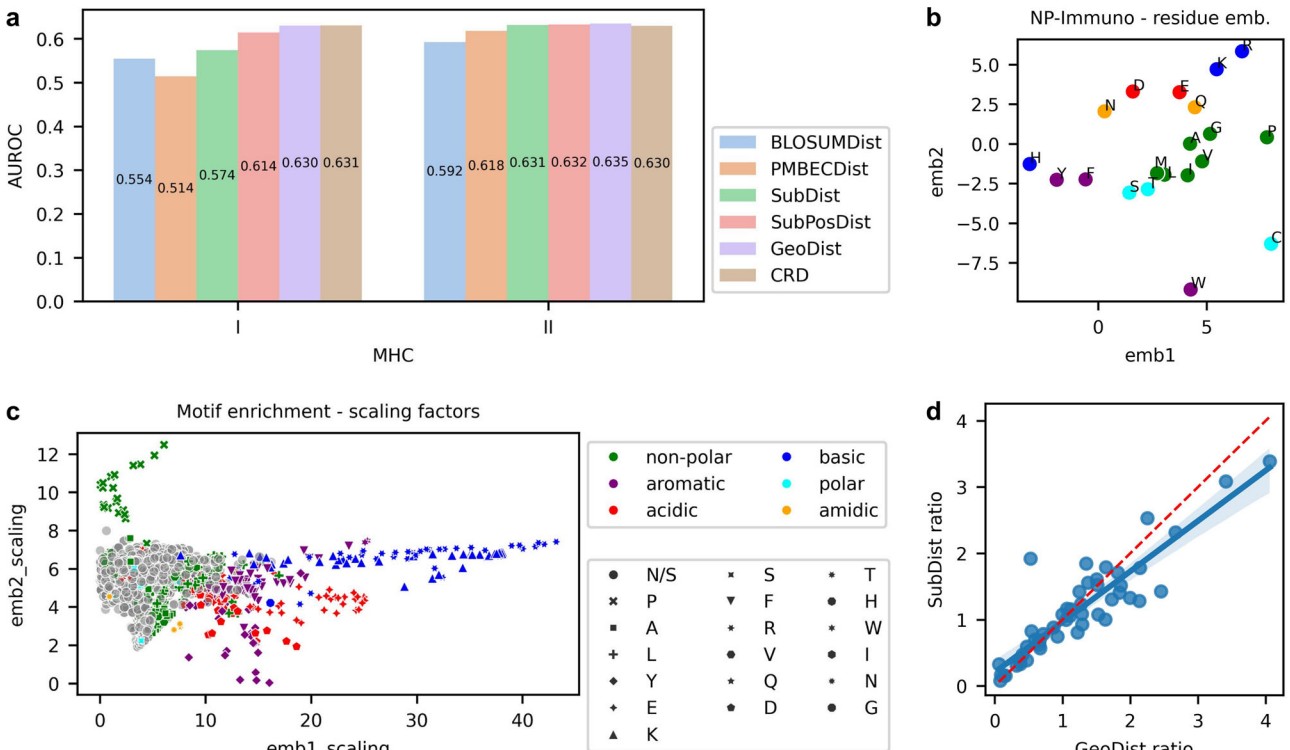

**Fig. 3 | Interpretation of NeoPrecis-Immuno. a** AUROC comparisons of incrementally constructed NeoPrecis-Immuno components for MHC-I and MHC-II predictions. Each component builds upon the previous one, except for BLOSUMDist and PMBECDist, which serves as baseline comparison. The components include: BLOSUMDist (BLOSUM62 distance, baseline), PMBECDist (PMBEC distance, baseline), SubDist (substitution distance of residue embeddings), SubPosDist (SubDist with position weighting), GeoDist (SubPosDist with MHC-binding motif enrichment), and CRD (GeoDist with sigmoid scaling). **b** Distribution of residue embeddings in NeoPrecis-Immuno. Residues are colored by amino acid properties to illustrate clustering based on biochemical characteristics. **c** Scaling factors for

motif enrichment across MHC allele-position pairs ($n = 4923$). These factors indicate how motif enrichment adjusts residue embeddings (**b**) along specific axes. Each dot is annotated with the most frequently observed amino acid at the binding motif for that allele position, represented by both color and text. **d** Positive-to-negative ratio distributions for SubDist and GeoDist across MHC-I allele-position pairs ($n = 50$) in the NCI dataset. Positives represent immunogenic substitutions while negatives represent non-immunogenic substitutions. A higher ratio indicates better differentiation. The red line represents x = y, while the blue line shows the fitted regression; the shaded region represents the 95% confidence interval.

contributed minimally to CRD, as GeoDist and CRD scores were highly correlated (Supplementary Fig. 4a).

Plotting amino acid embeddings on a 2D projection, NeoPrecis-Immuno residue embeddings grouped amino acids based on their chemical properties, with similar residues clustering together (Fig. 3b). Notably, NeoPrecis-Immuno embeddings differed from BLOSUM62 embeddings; for instance, histidine (H) was positioned closer to tyrosine (Y) and phenylalanine (F), likely due to their shared aromatic rings, while tryptophan (W) was positioned further away, possibly reflecting its larger size (Supplementary Fig. 4c). Thus, NeoPrecis-Immuno embeddings obtained through training on TCR-pMHC interaction data differ from BLOSUM62 embeddings, derived from evolutionary conservation more broadly.

Position factors revealed that the third and fifth residue positions were most influential for MHC-I and MHC-II, respectively (Supplementary Fig. 4b). Interestingly, these are not typical anchor positions for MHC binding, suggesting their significance lies in TCR-pMHC interactions. This hypothesis is supported by the cross-reactive triplet dataset, where amino acid substitutions with significant property changes (BLOSUM62 substitution score ≤ 0) at these positions were rare (Supplementary Fig. 1b).

The motif enrichment step of NeoPrecis-Immuno played a critical role in refining residue embeddings. Using affine transformations (rotation, reflection, and scaling) for interpretation, we analyzed how motif-enriched residue embeddings align with MHC-binding motifs (Methods). Among these transformations, scaling was the primary contributor to differences between wild-type and mutated amino

acids, with scaling factors being specific to each position and allele. This specificity likely reflects the variation in MHC-binding motifs across different positions. Positions dominated by the same amino acid exhibited consistent scaling behavior. For example, positions dominated by proline tended to scale the y-axis, while those dominated by arginine scaled the x-axis (Fig. 3c). These refinements adjusted the relative distances between amino acids in the embedding space, improving the ability of NeoPrecis-Immuno to differentiate wild-type and mutated residues for immunogenic neoantigen prediction.

To quantify the impact of motif enrichment, we analyzed the positive-to-negative distance ratio for different metrics, where positive distance indicates the distance between a wild-type peptide and its immunogenic mutated counterpart, while negative distance indicates the distance between a wild-type peptide and its non-immunogenic mutated counterpart. Neoantigens were grouped by allele and position to minimize bias, and the mean distances for positive and negative pairs were used to compute the ratio. GeoDist demonstrated a larger positive-to-negative ratio compared to SubDist (Fig. 3d), revealing that motif enrichment enhanced the model's ability to capture meaningful distance relationships between amino acids and better distinguish immunogenic from non-immunogenic samples.

We further computed an allele benefit score by averaging scaling factors across positions by MHC molecule (Supplementary Data 4, Methods). A higher benefit score indicates greater overall scaling of amino acid distance, reflecting increased immunogenicity and potential benefits for tumor immunity. To analyze trends, we categorized

alleles based on the most dominant amino acid at the anchor position of their binding peptides, which contributes most to the allele benefit score. MHC-I alleles associated with glutamic acid (E), lysine (K), or arginine (R) exhibited the highest benefit scores (Supplementary Fig. 4d), aligning with previously reported benefit HLA supertypes, including HLA-B27, which predominantly binds peptides with arginine at the anchor position, and HLA-B44, which favors peptides with glutamic acid[56,57].

## Multi-dimensional metrics for neoantigen immunogenicity

Beyond TCR recognition, neoantigen abundance and MHC presentation are two major factors contributing to immunogenicity. We defined eight key features across the three dimensions and evaluated these dimensions with the NCI dataset[54] (Methods). For neoantigen abundance, DNA allele frequency (AF) serves as a rough approximation of mutation clonality[58], while RNA AF and RNA expression levels quantify the expression of mutated peptides. To capture MHC presentation, patient harmonic-mean best rank (PHBR) measures binding affinity across an individual's MHC alleles[59], and robustness indicates the number of binding MHC alleles. For TCR recognition, in addition to the NeoPrecis-Immuno score, we included agretopicity ratio, proposed by Duan et al.[25], and foreignness, proposed by Łuksza et al.[26].

While PHBR, which aggregates NetMHCpan predictions across MHC alleles using a harmonic mean, has shown robust mutation-centric performance for predicting cell surface presentation[59], no standard method exists for aggregating TCR recognition metrics across alleles. To address this, we tested several aggregation strategies (Supplementary Fig. 5a, b, Methods); The masked maximum approach, which prioritizes the peptide with the highest recognition score across alleles that meet a minimum binding threshold necessary for presentation, either outperformed or matched the best binding approach across all three recognition metrics when evaluated on T-cell activity (Fig. 4a). Unlike the best binding approach, which focuses solely on the MHC allele with the highest binding prediction, the masked maximum approach considers all potential binding alleles. These results suggest that TCR recognition is best captured by considering all alleles capable of effective presentation, rather than focusing solely on the strongest binding peptide.

We next assessed the association between mutation-centric features and T-cell activity across all missense mutations ($n$ = 6952; CD8+ positive = 56; CD4+ positive = 66). Consistent with established immune mechanisms, MHC-I metrics were primarily associated with CD8 + T-cell activity, while MHC-II metrics correlated with CD4 + T-cell activity (Fig. 4b). Among the metrics, RNA AF, RNA expression, PHBR, and NeoPrecis-Immuno emerged as the strongest predictors of immunogenicity. DNA AF showed particular importance for MHC-I, while agretopicity ratio and foreignness were more influential for MHC-II predictions.

To refine predictions, we integrated the mutation-centric features into a logistic regression model. After performing feature selection, abundance features (A): DNA AF, RNA AF, and RNA expression; presentation feature (P): PHBR; and recognition feature (R): NeoPrecis-Immuno were included in the final integrated multi-dimensional model, NeoPrecis-Integrated (Supplementary Fig. 6a, Methods). Cross-validation repeated 100 times on the NCI dataset demonstrated that the A + P + R and A + R achieved similar performance (average AUROC: 0.848 vs. 0.849 for MHC-I; 0.806 for MHC-II), both substantially outperforming single-feature models (Fig. 4c, Supplementary Fig. 6b). This occurs because NeoPrecis-Immuno already incorporates MHC binding predictions as a covariate, effectively capturing presentation information. Feature importance analysis confirmed that recognition dominates in A + P + R while presentation contributes minimally, whereas presentation becomes important in A + P models (Fig. 4d, Supplementary Fig. 6c). These results demonstrate that integrating multi-dimensional features improves

immunogenicity prediction compared to single features alone, with NeoPrecis-Immuno serving as a comprehensive metric that captures both T-cell recognition and presentation information.

## Advancing immunotherapy response analysis with NeoPrecis

TMB and TNB are designed to quantify the number of potentially immunogenic neoantigens, but often underperform due to their limited consideration of immunogenic factors. To enhance neoantigen identification, we applied NeoPrecis to a meta ICI cohort comprising five melanoma[60–64] and three non-small cell lung cancer (NSCLC)[35,65,66] cohorts (Supplementary Table 2, Methods). This dataset included 695 samples with whole-exome sequencing data. We filtered for patients with pre-treatment biopsies and no prior ICI treatment to avoid pre-existing immune effects (Fig. 5a). One sample that failed clonal analysis and nine samples lacking RECIST labels were excluded, resulting in 525 patients (277 melanoma and 248 NSCLC). For subsequent analyses, ICI response was defined as complete response (CR) and partial response (PR), while non-response was defined as stable disease (SD) and progressive disease (PD). Overall survival (OS) was assessed for melanoma, while progression-free survival (PFS) was evaluated for NSCLC due to data availability (Supplementary Data 5). To assess cohort compatibility, we evaluated TMB distributions and mutation profiles (Methods). Pairwise Mann-Whitney U tests with FDR correction identified significant TMB differences in only a few cohort pairs (Supplementary Fig. 7a). Principal component analysis of binary mutation profiles revealed no obvious cohort-specific clustering, with silhouette scores of -0.36 for melanoma and -0.30 for NSCLC indicating minimal cohort-driven structure (Supplementary Fig. 7b).

First, we assessed the allele-level contribution to ICI treatment outcomes using the allele benefit score, derived from the NeoPrecis-Immuno model, which quantifies how MHC alleles refine neoantigen immunogenicity. A patient-centric allele benefit score (BenefitScore) was calculated as the geometric mean across all alleles (Methods). We then applied a Cox proportional hazards model, incorporating age and sex as confounders, to estimate its prognostic value. BenefitScore-dual, which integrates both MHC-I and MHC-II, was statistically significant in predicting patient outcomes for both melanoma ($p$-value = 0.04) and NSCLC ($p$-value = 0.01), and outperformed BenefitScore-I (MHC-I only) and BenefitScore-II (MHC-II only) (Fig. 5b). We further compared its performance against TMB as a baseline. While Benefit-Score alone did not surpass TMB, their combination (CombinedScore) improved the overall predictive power (Methods). These findings support that different MHC alleles modulate neoantigen immunogenicity in distinct ways, ultimately influencing ICI treatment outcomes.

Since allele-level assessment does not directly account for mutations, we next evaluated the mutation-level contribution using NeoPrecis-Immuno burden (NPB), defined as the number of mutations passing an NeoPrecis-Immuno threshold of 0.4 (approximately the median) for both MHC-I and MHC-II (Methods). NPB demonstrated superior performance in stratifying immune responses compared to TMB and TNB, as evidenced by both Cliff's Delta effect size (Fig. 5c) and a non-parametric statistical test (Supplementary Fig. 8a), suggesting that NeoPrecis-Immuno improves neoantigen immunogenicity predictions by accounting for TCR recognition.

To conduct a more comprehensive mutation-level evaluation, we calculated a tumor-centric immunogenicity score, NeoPrecis-Land-scapeSum, by summing NeoPrecis-Immuno predictions across all mutations (Methods). For each mutation, the MHC-dual NeoPrecis-Immuno score was derived by multiplying the MHC-I and MHC-II NeoPrecis-Immuno predictions. This multiplication strategy prioritizes mutations capable of activating both CD4+ and CD8 + T-cell responses. Effective anti-tumor immunity requires coordinated CD4 + T-cell, CD8 + T-cell, and dendritic cell responses, which depend on presenting both MHC-I and MHC-II neoantigens within the same antigen-

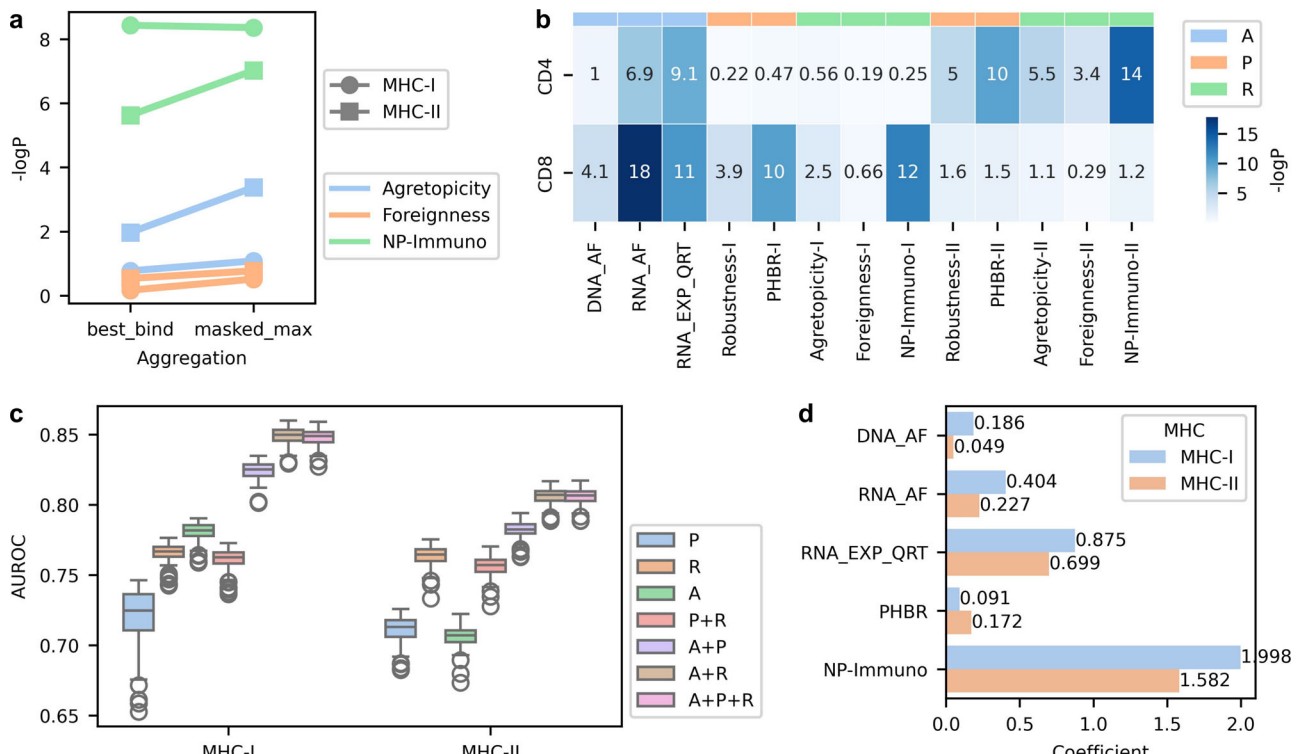

**Fig. 4 | Integration of multi-dimensional metrics for immunogenicity prediction. a** Comparison of allele aggregation methods. Results of the one-sided Mann-Whitney U test assess the association between TCR recognition metrics and T cell activities. The number of peptides was $n = 1448$ for MHC-I and $n = 1373$ for MHC-II. "Best_bind" indicates the best-binding allele approach, while "masked_max" refers to the masked maximum approach. **b** Association test for individual metrics in the NCI cohort ($n = 6952$ peptides). One-sided Mann-Whitney U test results evaluate the relationship between each metric and T cell activities, where "NP" denotes "Neo-Precis", "I" represents MHC-I, "II" represents MHC-II, "A" denotes abundance metrics, "P" denotes presentation metrics, and "R" denotes recognition metrics.

"RNA_EXP_QRT" denotes the quartile of RNA expression level (TPM). Agretopicity denotes the agretopicity ratio. **c** AUROC values for 4-fold cross-validation repeated 100 times on the NCI cohort using different combinations of mutation-centric features ($n = 6952$; CD8+ positive = 56 for MHC-I; CD4+ positive = 66 for MHC-II). In the boxplots, the center line represents the median of 100 repeats, boxes indicate the interquartile range (IQR; 25th–75th percentile), and whiskers extend to the most extreme data points within 1.5× the IQR. Outliers are depicted as individual circles. **d** Feature importance in the logistic regression model integrating multi-dimensional metrics. Features were standardized prior to modeling, and importance values are derived from logistic regression coefficients.

presenting cell[43]. Given intratumoral heterogeneity, a single mutation activating both pathways is more likely to elicit coordinated immunity than relying on separate mutations that may not co-occur in the same cell. We compared multiplication (I×II) with summation (I + II) and maximum (max(I, II)) aggregation methods. While all methods showed comparable performance at the mutation level (Supplementary Fig. 8b), multiplication was selected based on its superior performance at the subclone level (see below, Supplementary Fig. 10). This integrated approach demonstrated superior performance over TMB and TNB in both melanoma (AUROC = 0.66) and NSCLC (AUROC = 0.75) (Fig. 5d). Given the better performance of MHC-dual multiplication, we refer to the tumor-centric immunogenicity score as MHC-dual multiplication in subsequent analyses. In survival analyses (Methods), patients in the top half of the tumor-centric score showed significantly better outcomes than those in the bottom half (Fig. 5e). NeoPrecis-LandscapeSum outperformed TMB in melanoma and showed comparable performance in NSCLC (Supplementary Fig. 8c).

Lastly, we evaluated the efficacy of the multi-dimensional integrated immunogenicity prediction model (NeoPrecis-Integrated) in patients with RNA-seq data, identifying 187 samples (137 melanoma and 50 NSCLC). The NeoPrecis-Integrated model, trained on the NCI cohort using five selected features (Methods), demonstrated improved performance over NeoPrecis-Immuno in melanoma but not in NSCLC (Supplementary Fig. 9a). Here, the lower purity of NSCLC tumors (Supplementary Fig. 9b) may have impacted the inference of neoantigen abundance metrics, such as DNA AF and RNA AF, implicating the importance of data quality for robust predictions.

## Clonality-aware neoantigen landscapes in tumor heterogeneity

Mutation burden or summation metrics cannot account for the distribution of mutations within tumors. To address this limitation, we sought to characterize the tumor-specific neoantigen landscape by incorporating both immunogenicity and tumor subclonal architecture. First, we performed clonal analysis using PyClone[67], grouping mutations into clusters annotated with their prevalence in the tumor and mutation number (Methods). We developed two approaches to integrate tumor clonality: NeoPrecis-LandscapeCCF, which calculates a weighted sum of NeoPrecis-Immuno scores using cancer cell fraction (CCF), and NeoPrecis-LandscapeClone, which aggregates NeoPrecis-Immuno scores by cluster architecture (Methods). For the latter, the NeoPrecis-LandscapeClone scores of mutations within a cluster were summed to determine cluster immunogenicity, which was then averaged across clusters weighted by prevalence to derive tumor-centric immunogenicity (Fig. 6a).

We evaluated the same three MHC-I and MHC-II integration strategies (multiplication, summation, maximum) at the subclone level, where subclone-level MHC-I and MHC-II scores were calculated separately and then integrated (Methods). Multiplication consistently outperformed other methods in both melanoma and NSCLC (Supplementary Fig. 10). This superior performance at the subclone level provides additional biological support for the coordination between CD4+ T cell, CD8+ T cell, and dendritic cell. Since tumor cells within the same subclone share mutations, dendritic cells in that subclone can present multiple neoantigens on both MHC-I and MHC-II, enabling coordinated dual-class T-cell responses against the same tumor subclone.

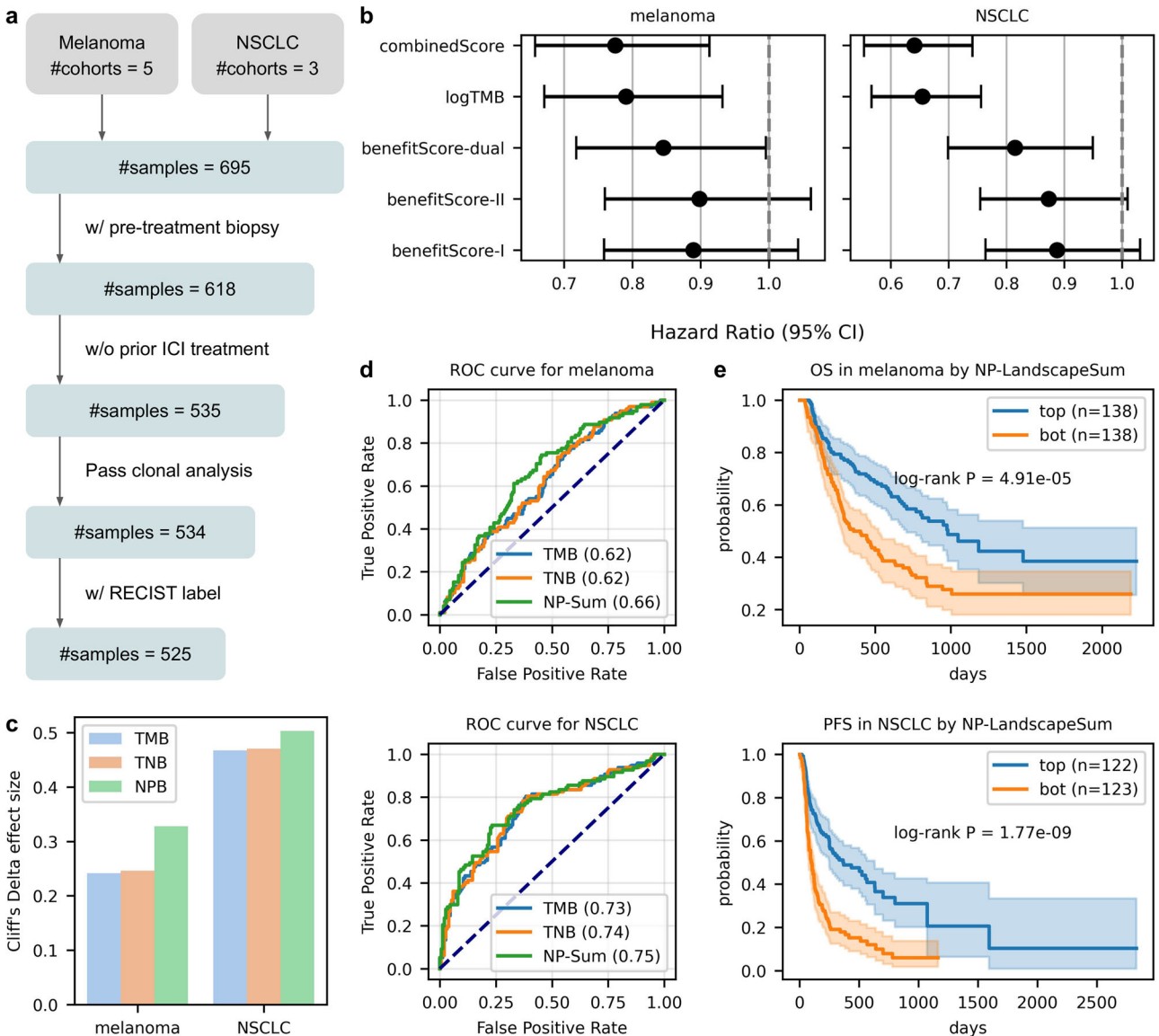

**Fig. 5 | Prediction of ICI response using the neoantigen landscape model.**
**a** Criteria for patient inclusion in the analysis. **b** Hazard ratio of metrics related to allele benefit scores derived from the Cox proportional hazards model with sex and age as confounders. Overall survival (OS) is used for melanoma ($n = 276$ patients), while progression-free survival (PFS) is used for NSCLC ($n = 245$). "Dual" is the metric considering both MHC-I and MHC-II. "CombinedScore" is the product of "logTMB" and "BenefitScore-dual". Data are presented as the estimated Hazard Ratio (center dot) and 95% confidence interval (error bars). **c** Cliff's Delta effect sizes comparing the impact of different mutation burdens on ICI response,

including tumor mutation burden (TMB), tumor neoantigen burden (TNB), and NeoPrecis-Immuno burden (NPB), for melanoma ($n = 277$; #positives = 98) and NSCLC ($n = 248$; #positives = 97). **d** AUROC comparison of ICI response prediction using TMB, TNB, and NeoPrecis-LandscapeSum (NP-Sum) in melanoma and NSCLC. **e** Kaplan-Meier survival curves stratified by NP-LandscapeSum, with log-rank P-values for melanoma (OS) and NSCLC (PFS). "Top" refers to the top half of patients with higher NP-LandscapeSum scores, while "bot" refers to the bottom half of patients with lower NP-LandscapeSum scores. Shaded areas indicate the 95% confidence interval. NP denotes NeoPrecis.

We benchmarked NeoPrecis against existing approaches that also incorporate mutation clonality within tumors, including the Cauchy-Schwarz index of neoantigens (CSiN)[68] and the immunoediting-based optimized tumor neoantigen load (ioTNL)[69]. CSiN profiles the neoantigen landscape by weighting the neoantigen load of each mutation with its variant allele frequency, while ioTNL aggregates weighted neoantigen loads across subclones potentially subject to immune elimination (Methods). Overall, NeoPrecis outperformed other approaches by more effectively qualifying the immunogenicity of neoantigens within subclones, rather than simply quantifying their number (Supplementary Fig. 11a). To ensure this superior performance was not driven by cohort-specific effects, we assessed models within individual cohorts. NeoPrecis-LandscapeClone outperformed TMB in 4 of 5 melanoma cohorts, while NeoPrecis-LandscapeSum

outperformed TMB in the Ravi NSCLC cohort (Supplementary Fig. 11b). When incorporating cohort as a covariate in cross-validation, the relative performance of all approaches remained unchanged (Supplementary Fig. 11c), confirming the superiority of NeoPrecis across cohorts. In melanoma, our clonality-aware methods generally showed better performance, with NeoPrecis-LandscapeClone achieving the best AUROC of 0.69. However, in NSCLC, mutation burden-based methods yielded better results, with NeoPrecis-LandscapeSum demonstrating the top AUROC of 0.75 (Supplementary Fig. 11a).

To investigate the discrepancy between melanoma and NSCLC, we evaluated tumor heterogeneity for each sample. Tumor heterogeneity was quantified using the Gini index, which measures inequality, across both prevalence and size (mutation number) dimensions to better capture the distribution of mutation clusters (Methods). A high

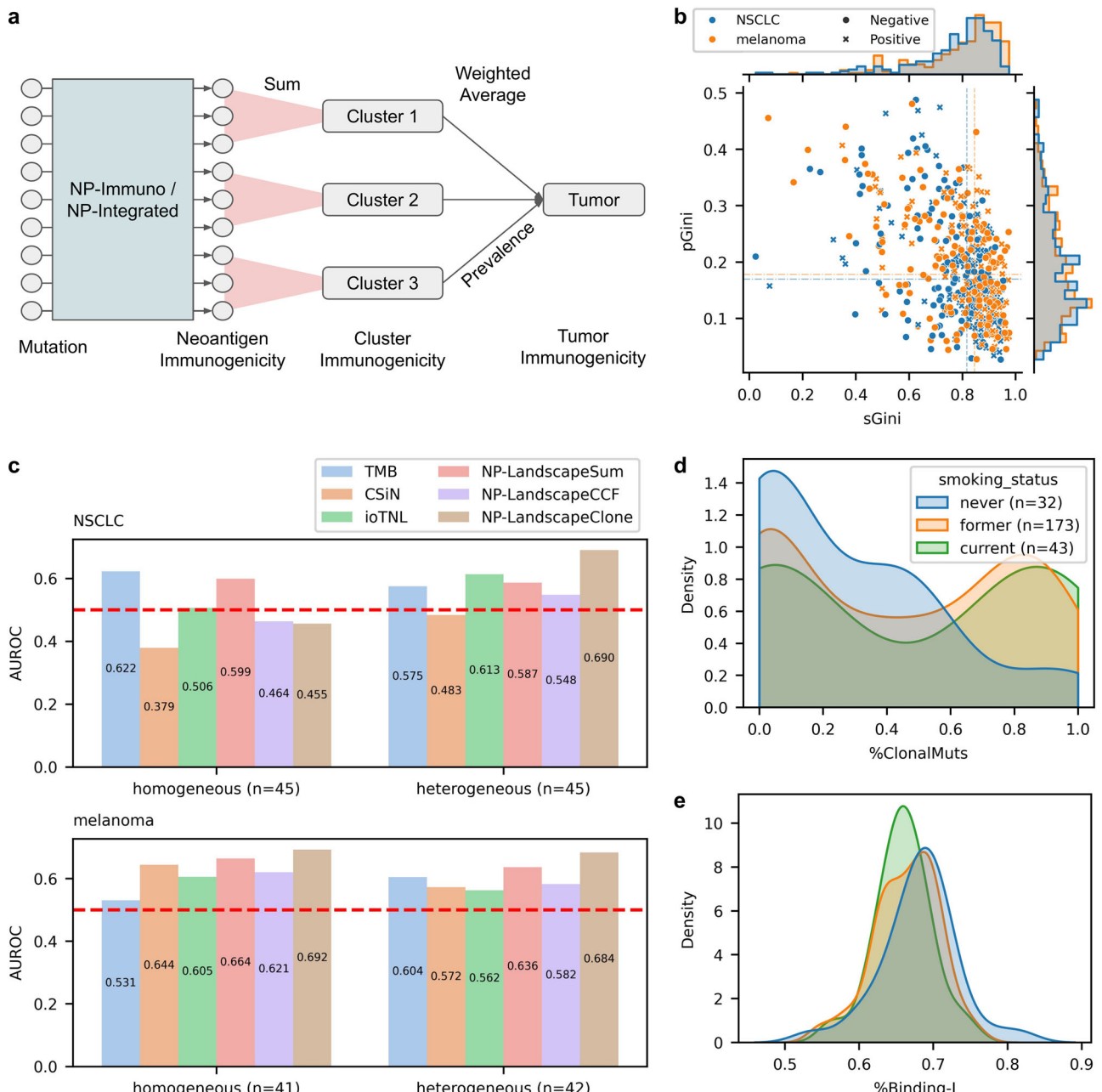

**Fig. 6 | Enhancing ICI prediction by incorporating tumor subclonal architecture. a** Computation of NeoPrecis-LandscapeClone. Immunogenicity predictions from NeoPrecis-Immuno or NeoPrecis-Integrated are summed within each mutation cluster. These cluster sums are then combined using a weighted average, where the prevalence of each cluster serves as the weight. This approach generates a tumor-centric immunogenicity score by integrating multiple mutation-centric predictions. **b** Distribution of mutation number, Gini index (sGini), and prevalence Gini index (pGini). ICI response (positive or negative) is annotated with the mark shape. **c** AUROC comparison of TMB, CSiN, ioTNL, NP-LandscapeSum, NP-

LandscapeCCF, and NP-LandscapeClone across two patient groups, heterogeneous (low sGini–low pGini) and homogeneous (high sGini–high pGini), in melanoma and NSCLC. NP denotes NeoPrecis. **d** Density plot estimated using kernel density estimation (KDE) showing the distribution of %ClonalMuts, the ratio of clonal mutations (CCF ≥ 0.85) to total mutations, across different smoking statuses. Values are clipped between 0 and 1. **e** Density plot estimated using KDE shows the distribution of %Binding-I, the ratio of MHC-I binding mutations (PHBR ≤ 2) to total mutations, across different smoking statuses. Values are clipped between 0 and 1.

Gini index indicates dominance of a few clusters and greater homogeneity. Conversely, a low Gini index reflects a more heterogeneous tumor. Surprisingly, the distributions of prevalence Gini index (pGini) and size Gini index (sGini) were similar between melanoma and NSCLC (Fig. 6b). Tumors were then classified based on their heterogeneity: the first quadrant (high pGini and high sGini) represented more homogeneous tumors, while the third quadrant (low pGini and low sGini) represented more heterogeneous tumors. In NSCLC, patients with heterogeneous tumors exhibited worse outcomes compared to those

with homogeneous tumors, although the difference was not statistically significant (Supplementary Fig. 12a). In contrast, no such trend was observed in melanoma. We further analyzed tumor heterogeneity in relation to smoking status in NSCLC. Never smokers had a higher proportion of heterogeneous tumors compared to smokers (Supplementary Fig. 13a). Additionally, they exhibited lower TMB (Supplementary Fig. 13b) and are known to have poorer responses to immunotherapy[70], aligning with the worse prognosis observed in the heterogenous group.

We evaluated the performance of the clonality-aware neoantigen landscape across the heterogeneity-defined groups (Methods, Fig. 6c, Supplementary Fig. 12b). NeoPrecis-LandscapeSum, which does not account for clonality, performed consistently across both groups and tumor types, indicating no subset bias between the groups. In melanoma, NeoPrecis-LandscapeClone consistently outperformed the other methods, regardless of tumor heterogeneity. In NSCLC, however, NeoPrecis-LandscapeClone showed superior performance in the heterogeneous group but underperformed in the homogeneous group. This pattern aligns with the weaker performance of NeoPrecis-LandscapeSum in never smokers (Supplementary Fig. 13c), who tend to have more heterogeneous tumors. These findings suggest that, as predicted, the clonality-aware neoantigen landscape is particularly effective at capturing immunogenic neoantigens in heterogeneous NSCLC tumors.

The clonality-aware neoantigen landscape primarily reflects the impact of clonal neoantigens; its poor performance in homogeneous NSCLC tumors suggests that clonal neoantigens may play a less significant role in these cases. One possible explanation is that these tumors have undergone stronger immunoediting, leading to the selective elimination of highly immunogenic clonal neoantigens. To investigate this, we examined the proportion of clonal mutations (CCF ≥ 0.85)[35] and the ratio of MHC-I binding mutations (PHBR ≤ 2) across different smoking statuses (Fig. 6d, e). Smokers exhibited a higher proportion of clonal mutations but a lower fraction of binding mutations, suggesting immunoediting or the intrinsic nature of smoking-induced mutagenesis yielding non-MHC binding peptides. This provides a potential explanation for the weaker performance of the clonality-aware neoantigen landscape in homogeneous NSCLC tumors, as the remaining clonal neoantigens may have survived due to low immunogenicity or immune evasion mechanisms, thereby weakening their predictive value.

## Discussion

Computational approaches to prioritize neoantigens remain an important area of development to provide biomarkers of immunotherapy response and provide insights into the characteristics of mutated peptides that govern immunogenicity[71,72]. Our immunogenicity model, NeoPrecis-Immuno, enhances predictive performance by incorporating MHC-binding motifs, refining how peptides are represented in relation to TCR recognition. The allele benefit score derived from this model further demonstrates significant predictive power for patient outcome in ICI treatment. In addition, we integrated both MHC-I and MHC-II predictions to improve ICI response prediction, recognizing the cooperative roles of CD8+ and CD4+ T cells in antitumor immunity. By further incorporating tumor clonality, we improved response prediction in melanoma and provided a more nuanced description of heterogeneous tumors in NSCLC.

Peptide binding to MHC molecules is a critical determinant of pMHC-TCR interactions, traditionally thought to be driven primarily by non-anchor residues that directly contact the TCR. However, emerging evidence suggests that MHC molecules influence TCR recognition beyond their role in peptide presentation. Wu et al. demonstrated that MHC polymorphisms can induce conformational changes in pMHC complexes, affecting TCR-pMHC interactions and modulating immune activation[73]. Additionally, modifications to MHC-binding anchor residues have been shown to alter TCR recognition. Cole et al. found that anchor-modified peptides could activate T cells with distinct TCRs[74], while Smith et al. showed that anchor residues function as allosteric modulators, shaping pMHC-TCR interactions[75]. These findings align with our model interpretation, where MHC-specific motif enrichment refines the embedding space at different scales, particularly capturing the high variability of anchor residues, ultimately improving neoantigen immunogenicity prediction.

Beyond amino acid substitutions and MHC molecules, which influence TCR recognition, we hypothesize that the entire peptide sequence also contributes to immunogenicity. Richman et al. demonstrated that dissimilarity to the human proteome correlates with peptide immunogenicity[76], while Łuksza et al. introduced a "non-selfness" metric based on neoantigen similarity to known antigens[31]. In our model, we attempted to capture peptide-level immunogenicity using a sigmoid-scaled cross-reactivity distance, allowing the model to learn how peptide sequence influences immune recognition. However, component contribution analysis showed that the sigmoid scaling factor had minimal impact on prediction performance (Fig. 3a), likely due to the limited sample size of the training set. The human proteome consists of millions of peptides, yet our model was trained on only thousands, potentially restricting its ability to distinguish self from non-self effectively. More targeted modeling of peptide selfness may be necessary to further enhance immunogenicity predictions.

In the context of ICI response evaluation, while MHC-I-restricted neoantigens have traditionally been the primary focus in mutation burden analysis, growing evidence underscores the essential role of MHC-II in tumor immunity. Magen et al. demonstrated that ICI response correlates with the clonal expansion of both intratumoral CD4+ and CD8+ T cells in hepatocellular carcinoma patients[77]. Similarly, Espinosa-Carrasco et al. emphasized the necessity of the immune triad—CD4+ T cells, CD8+ T cells, and dendritic cells—for effective tumor elimination[43]. These findings align with our results, showing that combining MHC-I and MHC-II predictions improves ICI response prediction at the subclone level (coordinated immune responses against tumor subclones presenting multiple neoantigens on both MHC classes; Supplementary Fig. 10) and shows comparable performance at the mutation level (single mutations activating dual pathways; Fig. 5b, Supplementary Fig. 8b).

Tumor heterogeneity remains a major obstacle to effective ICI responses[34–38], underscoring the need to identify patients with heterogeneous tumors who may derive therapeutic benefit. The NeoPrecis clonality-aware neoantigen landscape, NeoPrecis-LandscapeClone, significantly improved response prediction in heterogeneous NSCLC tumors. However, its performance was notably weaker in homogeneous NSCLC tumors, where dominant clones with a lower proportion of binding mutations likely underwent immunoediting, reducing their impact on immunogenicity (Fig. 6d, e). This is supported by Rosenthal et al.[78], who demonstrated that clonal mutations in early-stage NSCLC may undergo immunoediting, suggesting their role in shaping response to ICI. Nevertheless, this phenomenon is not observed in melanoma, which has a higher ratio of clonal mutations and a greater proportion of binding mutations (Supplementary Fig. 14a, b). Niknafs et al.[79] indicated key differences in clonal persistent TMB between melanoma and NSCLC, showing that melanoma exhibits a stronger correlation between immune response and clonal persistent TMB, aligned with our observations. These findings suggest that tumor type-specific adaptations are necessary to optimize neoantigen landscape modeling.

While NeoPrecis significantly enhances the characterization of neoantigen immunogenicity, it has certain limitations. First, the model is currently not applicable to indels or frameshift mutations, as it requires a defined wild-type peptide sequence. However, these mutations can generate highly immunogenic neoantigens and should be considered in future extensions[80]. Additionally, the model focuses solely on the core binding region, whereas MHC-II has been shown to involve flanking regions that contribute to both binding affinity and immunogenicity[81,82]. Furthermore, key immunogenic determinants, including germline variations, the tumor microenvironment, and T-cell infiltration, are not explicitly modeled. Sayaman et al.[83] and Pagadala et al.[84] demonstrated that germline predisposition variants can shape the tumor microenvironment, affecting immune infiltration and, consequently, tumor immunity. Future efforts should integrate these immune and tumor-intrinsic factors to further refine neoantigen prediction and improve personalized immunotherapy strategies.

## Methods

### Preparation of the cross-reactive peptide triplet dataset

TCR-peptide binding data were obtained from IEDB[48] and VDJdb[49]. Binding complexes were included if they contained a complete annotation of the beta complementarity-determining region 3 (CDR3), peptide sequence, and MHC allele. To isolate TCR recognition from the effects of MHC binding, peptides with low MHC-binding affinity were removed using a %rank threshold of 2 for MHC-I and 10 for MHC-II, based on predictions from NetMHCpan-4.1[52] (MHC-I) and NetMHCIIpan-4.3[53] (MHC-II). After filtering, the dataset included 69,278 complexes for MHC-I and 3165 complexes for MHC-II. Peptide-MHC complexes were then grouped by MHC allele and CDR3 sequence, ensuring that all peptides within a group bound to the same MHC and TCR molecule. Groups containing only one peptide were removed.

To train the immunogenicity model to distinguish mutated peptides from wild-type peptides, we constructed a dataset of cross-reactive peptide triplets. Each triplet consists of a seed peptide (a reference peptide), a cross-reactive peptide (sharing TCR binding with the seed), and a non-cross-reactive peptide (not sharing TCR binding) (Supplementary Fig. 1a). The model was trained to minimize the distance between seed-cross-reactive pairs (similar peptides) while maximizing the distance between seed-non-cross-reactive pairs (dissimilar peptides). To mimic the effect of single amino acid substitutions, the difference between both cross-reactive and non-cross-reactive pairs was constrained to one Hamming distance. Cross-reactive pairs were sampled from each peptide-MHC complex group to ensure functional similarity between the seed and cross-reactive peptides. All possible pairs were sampled, yielding 1153 MHC-I and 261 MHC-II cross-reactive pairs.

For each cross-reactive pair, non-cross-reactive peptides were randomly selected to be one Hamming distance away from the seed but absent from the cross-reactive set, which comprised peptides within the same peptide-MHC complex group. A BLOSUM62 matrix was used as a prior, ensuring that the substitution between seed and non-cross-reactive peptides resulted in a negative BLOSUM62 score, indicating substantial sequence dissimilarity. To enhance dataset diversity, we applied 10-fold generation, ultimately producing 11,530 MHC-I and 2610 MHC-II triplets (Supplementary Data 1).

### Preparation of the CEDAR immunogenicity dataset

Cancer-related immunogenicity data were obtained from CEDAR. Samples were included if they contained annotations for the MHC allele, wild-type peptide, mutated peptide, and immunogenicity label (T-cell assay). To ensure accurate modeling of substitutions, samples where the wild-type and mutated peptides had mismatched lengths were removed. To isolate TCR recognition from the effects of MHC binding, peptides with low binding affinity were filtered out using a % rank threshold of 2 for MHC-I and 10 for MHC-II. After filtering, the dataset comprised 4176 MHC-I samples and 87 MHC-II samples (Supplementary Data 2). Then, the dataset was split into training (75%), validation (10%), and testing (15%) sets. The testing set initially contained 640 samples, from which 438 unique peptides—ensured to be unobserved by any of the predictors—were retained for evaluation. Due to the limited number of MHC-II samples, this restriction was not applied to MHC-II.

### Preparation of the NCI cohort

The NCI gastrointestinal cancer cohort, derived from Parkhurst et al.[54], was utilized to evaluate the immunogenicity model and construct a multi-dimensional integrated model. The cohort comprises 6952 missense mutations across 75 patients, with both MHC-I and MHC-II alleles available for all patients, along with abundance data (DNA allelic fraction, RNA allelic fraction, and RNA expression level). Notably, RNA expression levels were quartile-normalized for each individual. Each

mutated peptide underwent T-cell assay testing, resulting in mutation-centric data with labels for each mutation. This approach differs from peptide-centric data (e.g., CEDAR dataset) where each peptide has a label. In total, 56 mutations exhibited CD8+ activity, and 66 showed CD4+ activity. For peptide-centric features, such as MHC-binding affinity and immunogenicity scores, an aggregation approach was applied to obtain mutation-centric predictions. This method will be described in detail in a subsequent section.

To evaluate immunogenicity prediction, mutations were filtered based on abundance and MHC presentation metrics to exclude factors that could bias TCR recognition comparison. Mutations with zero DNA allelic fraction, zero RNA allelic fraction, or no expression level were removed. Additionally, mutations were filtered using a %rank threshold of 2 for MHC-I and 10 for MHC-II. The final dataset for immunogenicity prediction evaluation consisted of 1089 mutations for MHC-I (36 positives) and 1189 mutations for MHC-II (33 positives). For the construction and evaluation of the multi-dimensional integrated model, all 6952 missense mutations were included to account for all three dimensions of the analysis (Supplementary Data 3).

### NeoPrecis immunogenicity model

The NeoPrecis immunogenicity model, NeoPrecis-Immuno, is a custom machine learning framework that integrates key immunogenicity determinants, including residue-level embedding, MHC-binding motif enrichment, peptide embedding, geometric representation, and sequence-aware scaling (Fig. 2a).

**Input preparation.** pMHC complexes are first predicted using an MHC-binding predictor, and the 9-mer core binding region is identified as the input sequence. For each neoantigen, we obtain paired wild-type (WT) and mutant (MT) peptide sequences represented as one-hot encoded matrices: $\mathbf{P^{WT}}, \mathbf{P^{MT}} \in \mathbb{R}^{L \times 20}$, where $L = 9$ is the core binding region length and each row is a one-hot vector over 20 amino acids.

**MHC binding motif representation.** To represent MHC allele in a generalizable manner, we constructed binding motifs from NetMHC-pan predictions. We randomly sampled 100,000 peptides from the human peptidome (9-mers for MHC-I, 15-mers for MHC-II) and predicted binding affinity for each available allele. Peptides passing the binding rank threshold (2 for MHC-I, 10 for MHC-II) were retained, and their 9-mer binding cores were used to compute position-specific amino acid frequencies. The resulting binding motif $\mathbf{M} \in \mathbb{R}^{L \times 20}$ serves as the MHC allele representation, where each position contains the frequency distribution over 20 amino acids.

**Amino acid embedding.** One-hot encoded amino acids are first transformed using the BLOSUM62 substitution matrix, then projected into a latent space via a single-layer neural network:

$$\mathbf{E^{WT}} = f_{embed}\left(\mathbf{P^{WT}}\right) \in \mathbb{R}^{L \times d_a} \tag{1}$$

$$\mathbf{E^{MT}} = f_{embed}\left(\mathbf{P^{MT}}\right) \in \mathbb{R}^{L \times d_a} \tag{2}$$

$$\mathbf{E^{motif}} = f_{embed}(\mathbf{M}) \in \mathbb{R}^{L \times d_a} \tag{3}$$

where $d_a = 2$ is the amino acid embedding dimension.

**Motif enrichment.** To capture the complex interactions between peptide sequences and MHC alleles, we introduce a motif-dependent convolutional layer. The model learns $K = 8$ convolutional kernels

$\mathbf{W_k^{conv}} \in \mathbb{R}^{d_a \times d_a}$, which are modulated by the MHC motif embedding:

$$\mathbf{C_k} = \mathbf{E^{motif}}\mathbf{W_k^{conv}} \in \mathbb{R}^{L \times d_a}, k = 1, \ldots, K \quad (4)$$

These motif-enriched kernels are then applied position-wise to peptide embeddings:

$$\mathbf{H_k^{WT}} = \mathbf{C_k} \odot \mathbf{E^{WT}} \in \mathbb{R}^L \quad (5)$$

$$\mathbf{H_k^{MT}} = \mathbf{C_k} \odot \mathbf{E^{MT}} \in \mathbb{R}^L \quad (6)$$

where $\odot$ denotes element-wise multiplication followed by summation across the embedding dimension. The $K$ kernel responses are concatenated and projected to the peptide embedding dimension:

$$\mathbf{E_{enriched}^{WT}} = \left(\mathbf{W^{proj}}\left[\mathbf{H_1^{WT}}, \ldots, \mathbf{H_K^{WT}}\right]\right)^T \in \mathbb{R}^{L \times d_p} \quad (7)$$

$$\mathbf{E_{enriched}^{MT}} = \left(\mathbf{W^{proj}}\left[\mathbf{H_1^{MT}}, \ldots, \mathbf{H_K^{MT}}\right]\right)^T \in \mathbb{R}^{L \times d_p} \quad (8)$$

where $\mathbf{W^{proj}} \in \mathbb{R}^{d_p \times K}$ is a learnable projection matrix and $d_p = 2$ is the peptide embedding dimension.

**Position-weighted peptide pooling.** To aggregate position-enriched embeddings into fixed-length peptide representations, we employ position-weighted pooling. Each of the 9 core binding positions has a learnable weight $w_i$ (separate parameters for MHC-I and MHC-II):

$$\mathbf{v^{WT}} = \sum_{i=1}^{L} w_i \mathbf{E_{enriched, i}^{WT}} \in \mathbb{R}^{d_p} \quad (9)$$

$$\mathbf{v^{MT}} = \sum_{i=1}^{L} w_i \mathbf{E_{enriched, i}^{MT}} \in \mathbb{R}^{d_p} \quad (10)$$

where $\mathbf{E_{enriched, i}}$ denotes the embedding at position $i$. This design allows the model to emphasize structurally important positions in the MHC binding groove.

**Geometric representation.** To capture the distinction between wild-type and mutant peptides in a way that reflects TCR cross-reactivity, we construct a geometric representation in the peptide embedding space. The geometric transformation consists of three components:

$$\mathbf{v^{origin}} = \mathbf{v^{WT}} \in \mathbb{R}^{d_p} \quad (11)$$

$$\mathbf{v^{direction}} = \frac{\mathbf{v^{MT}} - \mathbf{v^{WT}}}{||\mathbf{v^{MT}} - \mathbf{v^{WT}}||_2 + \varepsilon} \in \mathbb{R}^{d_p} \quad (12)$$

$$d = ||\mathbf{v^{MT}} - \mathbf{v^{WT}}||_2 \in \mathbb{R} \quad (13)$$

where $\mathbf{v^{origin}}$ represents the wild-type peptide position, $\mathbf{v^{direction}}$ is the normalized direction of mutation in the embedding space, $d$ is the Euclidean distance between wild-type and mutant embeddings, and $\varepsilon$ is a small constant to prevent division by zero.

**Cross-reactivity distance (CRD).** To account for sequence context, we compute a scaling factor using a two-layer neural network:

$$s = f_{scale}\left(\left[\mathbf{v^{origin}}; \mathbf{v^{direction}}\right]\right) \quad (14)$$

The CRD is then computed as:

$$\mathrm{CRD} = d \cdot \sigma(s) \quad (15)$$

where $\sigma$ is the sigmoid function. Higher CRD values indicate lower predicted TCR cross-reactivity between wild-type and mutant peptides.

Immunogenicity prediction: The final immunogenicity prediction integrates CRD with MHC binding prediction:

$$y_{immuno} = \sigma\left(\mathbf{W_{out}}\left[\mathrm{CRD}; \mathrm{MHC_{binding}}\right] + b_{out}\right) \quad (16)$$

This produces an immunogenicity score between 0 and 1.

**Model training.** The NeoPrecis-Immuno model was trained in two stages. In the first stage, it was trained on the cross-reactive peptide triplet dataset using contrastive learning with triplet loss to optimize CRD:

$$\mathcal{L}_{triplet} = \max(0, \mathrm{CRD}(a, p) - \mathrm{CRD}(a, n) + \mathrm{margin}) \quad (17)$$

where $a$ represents a seed peptide, $p$ is a cross-reactive peptide, $n$ is a non-cross-reactive peptide, and the margin was set to 1. In the second stage, the model was fine-tuned on the CEDAR immunogenicity dataset using binary cross-entropy loss:

$$\mathcal{L}_{BCE} = -\frac{1}{N}\left[y_i \log(\hat{y}_i) + (1 - y_i) \log(1 - \hat{y}_i)\right] \quad (18)$$

where $y_i$ is the true immunogenicity label and $\hat{y}_i$ is the predicted score. During fine-tuning, all layers were frozen, except for the scaling factor and classifier, allowing the model to focus on adjusting immunogenicity predictions. Training was conducted for 100 epochs with a learning rate of 0.005, using the AdamW optimizer[85].

**Residue embeddings**
To interpret NeoPrecis, we utilized three distinct residue embeddings: BLOSUM62 embedding, NeoPrecis-Immuno residue embedding, and motif-enriched embedding. To generate BLOSUM62 embeddings, we applied principal component analysis (PCA) to the BLOSUM62 matrix, extracting the top two principal components as the embedding for each amino acid. NeoPrecis-Immuno residue embedding is also a two-dimensional representation designed to capture amino acid relationships in the context of TCR recognition. The motif-enriched embedding is a position-wise transformation of the NeoPrecis-Immuno residue embedding, incorporating position-specific MHC-binding motifs. Each position-specific motif is represented as a projection matrix, which is then multiplied with the residue embedding to refine its representation based on MHC-binding context. This transformation ensures that amino acid embeddings reflect not only intrinsic residue properties but also their contextual influence within the pMHC complex.

**Immunogenicity predictions from other predictors**
To benchmark NeoPrecis-Immuno, we evaluated its performance against three immunogenicity predictors: PRIME[30], ICERFIRE[27], and DeepNeo[28]. All three models provide MHC-I predictions, but only DeepNeo supports MHC-II. However, due to differences in MHC allele coverage, not all predictors were available for every allele in our datasets. To ensure fair comparisons in the CEDAR internal validation, we restricted the testing set to peptides supported by all predictors. For the NCI external validation, if a predictor did not support certain alleles in a patient, we removed those alleles and computed the mutation-centric immunogenicity score based on the remaining available alleles. Predictions were aggregated across alleles using the masked maximum approach. If a mutation lacked predictions from a given predictor across all its binding alleles, it was excluded from the testing set to maintain consistency in benchmarking.

For PRIME, we used PRIME 2.0, downloaded from GitHub. ICERFIRE was run as a stand-alone application, excluding expression values from its input. DeepNeo predictions were obtained using the DeepNeo-v2 web server, which only supports the beta chain of MHC-II (DQB, DPB). Consequently, alpha chains (DQA, DPA) were ignored when computing mutation-centric DeepNeo scores. These standardized procedures ensured a fair and unbiased evaluation of NeoPrecis-Immuno against existing immunogenicity predictors.

## Component contribution analysis

To assess the contribution of individual components in the NeoPrecis-Immuno, we systematically extracted and analyzed key features, including residue embedding, motif enrichment, position weighting, and CRD scaling factors, and compared them to baseline amino acid distance metrics.

For baseline comparison, we used BLOSUM62 and PMBEC[51] substitution matrices. BLOSUM62 is an amino acid substitution matrix derived from aligned protein sequences, while PMBEC is specifically derived from peptide-MHC binding data. Since substitution matrices provide discrete scores for amino acid pairs, we applied PCA to each matrix to extract the first two principal components as two-dimensional amino acid embeddings, enabling continuous distance calculations. We then computed distances as the Euclidean distance between wild-type and mutated amino acids in this reduced space (BLOSUM62Dist, PMBECDist).

NeoPrecis-Immuno progressively refines distance calculations through four stages. First, substitution distance (SubDist) uses trained residue embeddings to measure differences between wild-type and mutated peptides. This is then refined into position-weighted substitution distance (SubPosDist), which adjusts SubDist by incorporating positional importance, ensuring that residues within the peptide-binding core are weighted according to their functional contribution to TCR recognition. Further refinement is achieved through geometric distance (GeoDist), which extends SubPosDist by integrating motif enrichment. By incorporating MHC-binding motifs, GeoDist adjusts residue embeddings to better reflect TCR-pMHC interactions. Finally, cross-reactivity distance (CRD) applies a sigmoid scaling function to GeoDist, modifying geometric distances to account for sequence-specific variations in immunogenicity. This scaling ensures that even when geometric distances between wild-type and mutated peptides are similar, differences in sequence composition are considered in immunogenicity predictions.

## Affine transformation for the motif-enriched embedding

To interpret how motif enrichment refines residue embeddings, we applied an affine transformation between the original residue embedding and the motif-enriched embedding:

$$\mathbf{E_m} = \mathbf{AE_r} + \mathbf{t} \tag{19}$$

where $\mathbf{E_m}$ is the motif-enriched embedding, $\mathbf{E_r}$ is the original residue embedding, $\mathbf{A}$ is the transformation matrix, and $\mathbf{t}$ is the translation vector. To analyze how this transformation scales the embedding space, we performed singular value decomposition (SVD) on $\mathbf{A}$:

$$\mathbf{A} = \mathbf{U\Sigma V^T} \tag{20}$$

where $\mathbf{\Sigma}$ contains the scaling factors, and $\mathbf{V^T}$ provides the principal directions of transformation. The scaling effect along the original x- and y-axes was computed by projecting $\mathbf{\Sigma}$ onto the original coordinate system using $\mathbf{V^T}$:

$$S_x = |\mathbf{V_{1x}} \cdot \sigma_1| + |\mathbf{V_{2x}} \cdot \sigma_2|, \; S_y = |\mathbf{V_{1y}} \cdot \sigma_1| + |\mathbf{V_{2y}} \cdot \sigma_2| \tag{21}$$

where $S_x$ and $S_y$ represent the scaling along the x- and y-axes, and $\mathbf{v_1}$, $\mathbf{v_2}$ are the basis vectors from $\mathbf{V^T}$. This analysis quantifies how motif enrichment reshapes residue embeddings, refining their representation in the context of MHC-binding motifs.

## Allele benefit scores

Allele benefit scores are designed to evaluate how MHC-binding motifs refine neoantigen immunogenicity by influencing T-cell recognition. For each allele and position, we calculated scaling factors ($S_x$ and $S_y$), and the allele benefit score was defined as the average geometric mean of $S_x$ and $S_y$ across all 9-mer positions:

$$BenefitScore = \frac{1}{9}\sum_{i=1}^{9}\sqrt{S_{x,i} \cdot S_{y,i}} \tag{22}$$

A higher allele benefit score results from larger $S_x$ or $S_y$ values, indicating a greater distance between WT and MT peptides, which in turn suggests increased immunogenicity. Additionally, the allele benefit score can be aggregated across individual alleles to provide an overall assessment of an individual's immune response. The geometric mean was calculated separately for MHC-I and MHC-II alleles, where $N$ is the number of alleles:

$$MHC\ BenefitScore = \left(\prod_{j=1}^{N} BenefitScore_j\right)^{\frac{1}{N}} \tag{23}$$

The MHC-dual benefit score was computed as the geometric mean of the MHC-I and MHC-II benefit scores:

$$MHC - dual\ BenefitScore = \sqrt{(MHC - I\ BenefitScore) \cdot (MHC - II\ BenefitScore)} \tag{24}$$

To integrate with TMB, the combined score was calculated as the product of logTMB and MHC-dual benefit score:

$$CombinedScore = (\log TMB) \cdot (MHC - dual\ BenefitScore) \tag{25}$$

## Calculation of multi-dimensional metrics

Eight metrics were computed to evaluate neoantigen immunogenicity, categorized into abundance, presentation, and recognition metrics. Abundance metrics included DNA allele frequency (AF), RNA AF, and RNA expression quartile, while presentation metrics comprised robustness and PHBR. Recognition metrics included agretopicity ratio, foreignness, and NeoPrecis-Immuno. Abundance metrics were available in the NCI cohort, whereas for the Ott and ICI cohorts, DNA AF was derived from variant calling results, and RNA AF was calculated using bam-readcount[86]. RNA expression quartile was computed as the quartile transformation of TPM values obtained from RSEM[87] for each individual. Presentation metrics were determined as follows: robustness was defined as the number of binding alleles, using a %rank threshold of 2 for MHC-I and 10 for MHC-II. PHBR (patient harmonic-mean best rank) was computed as the harmonic mean of the %rank of the best-binding peptide for each allele. Recognition metrics included agretopicity ratio, which was calculated as the ratio of MHC binding scores between mutated and wild-type peptides, and foreignness, which measured sequence similarity to foreign immunogenic antigens. Foreignness was computed using the NeoantigenEditing repository (GitHub) by Łuksza et al.[31].

## T-cell recognition metric aggregation across MHC alleles

To predict mutation-level immunogenicity from allele-specific scores, we evaluated multiple aggregation strategies that account for individual MHC genotypes. Each strategy involves two key decisions: whether to apply allele masking and which aggregation method to use.

Allele masking: Masking excludes alleles predicted not to bind the target peptide, based on NetMHCpan %rank thresholds ($\leq 2$ for MHC-I, $\leq 10$ for MHC-II). Let $A_{all}$ denote all MHC alleles for an individual and $A_{bind} \subseteq A_{all}$ denote binding alleles. Without masking, all alleles contribute to aggregation; with masking, only $A_{bind}$ is considered.

Aggregation methods: For allele-specific scores $s_a$, we evaluated five aggregation methods.

- Mean: $s_{mut} = \frac{1}{|A|} \sum_{a \in A} s_a$
- Harmonic mean: $s_{mut} = \frac{|A|}{\sum_{a \in A}(s_a + \varepsilon)^{-1}}$
- Weighted average: $s_{mut} = \frac{\sum_{a \in A} w_a s_a}{\sum_{a \in A} w_a}$, where $w_a$ is binding affinity from NetMHCpan
- Best binding: $s_{mut} = s_{a^*}$, where $a^* = argmin_{a \in A} rank(a)$
- Maximum: $s_{mut} = \max_{a \in A} s_a$

We compared all ten combinations (with/without masking × five aggregation methods) on the NCI dataset and found that masked maximum approach performed most consistently across both MHC classes and different immunogenicity metrics (Supplementary Fig. 5b).

To assess the impact of MHC allele aggregation on immunogenicity prediction, we tested various approaches. The first consideration was masking, which excludes alleles that do not bind to the neoantigen based on the %rank threshold of 2 for MHC-I and 10 for MHC-II. The second consideration was the choice of aggregation method, where we evaluated mean, harmonic mean, weighted average (weighted by binding affinity), best binding, and maximum. For example, the masked maximum method selects the highest score among all binding alleles that meet the masking criteria. This approach ensures that only strongly binding alleles contribute to the aggregated immunogenicity score, refining the prediction of neoantigen immunogenicity.

## Feature selection for the integrated multi-dimensional model

To identify the most predictive features among the eight immunogenicity metrics, we applied recursive feature elimination with cross-validation (RFECV) using 4-fold cross-validation. Features were eliminated one at a time, and the subset yielding the best cross-validation performance was selected as the final feature set. For consistency, we retained five features for both MHC-I and MHC-II, even though the optimal feature set for MHC-II consisted of only two features. This decision was made because there was no significant difference in performance between using two and five features in MHC-II (Supplementary Fig. 6a), and the smaller sample size for MHC-II limited the reliability of feature selection.

## Integrated multi-dimensional model

The integrated model, NeoPrecis-Integrated, is a logistic regression trained on the selected features, with all features undergoing standard normalization based on the training set. The mean and standard deviation calculated from the training set were recorded and subsequently applied to the testing set for consistent normalization. To assess the impact of multi-dimensional integration, we conducted 4-fold cross-validation on the NCI dataset. For ICI response prediction, the integrated model was trained on the entire NCI dataset to ensure robust performance.

## Preparation of the ICI cohorts

We compiled eight patient cohorts treated with ICIs, including five melanoma cohorts and three NSCLC cohorts (Supplementary Table 2). In total, 695 patients (443 melanoma and 252 NSCLC) with whole-exome sequencing (WES) data were collected from dbGaP. Raw sequencing data were processed using a standardized pipeline. Variant calling was performed using Nextflow-Sarek (v3.4.2)[88,89], with the hg38 reference genome, employing Mutect2 for somatic variant calling, VEP for mutation annotation, and ASCAT for copy number analysis, tumor

purity, and ploidy estimation. RNA sequencing data were aligned to hg38 using STAR (v2.7.3a)[90] with default settings, and transcript quantification was conducted using RSEM (v1.3.1)[87]. HLA typing was performed with HLA-HD (v1.7.0)[91], ensuring accurate identification of patient-specific MHC alleles. Clonality analysis was conducted using PyClone (v0.13.1)[67], with 5,000 iterations and a maximum of 100 clusters, allowing for a detailed assessment of tumor heterogeneity.

## Calculation of mutation burden

We computed three distinct mutation burden metrics for comparison: tumor mutation burden (TMB), tumor neoantigen burden (TNB), and NeoPrecis-Immuno burden (NPB). TMB was defined as the total number of non-synonymous mutations in the tumor. TNB accounted for non-synonymous mutations that met an MHC-I binding threshold (PHBR $\leq 2$), incorporating MHC-binding constraints into the burden calculation. NPB incorporated both MHC-I and MHC-II binding along with TCR recognition. A mutation was counted if the product of NeoPrecis-Immuno-I (MHC-I) and NeoPrecis-Immuno-II (MHC-II) scores was $\geq 0.16$. This threshold was derived from the median NeoPrecis-Immuno score (around 0.4) for both MHC-I and MHC-II, where 0.16 represents the squared value (0.4 × 0.4), ensuring that mutations with strong immunogenic potential were captured.

## Calculation of tumor-centric immunogenicity score

Aggregating neoantigen immunogenicity scores into a tumor-centric immunogenicity score is essential for assessing individual immune response. To achieve this, we first computed mutation-centric immunogenicity scores using three distinct metrics: NeoPrecis-Immuno-I, representing the NeoPrecis-Immuno score for MHC-I; NeoPrecis-Immuno-II, representing the NeoPrecis-Immuno score for MHC-II; and NeoPrecis-Immuno-dual, calculated as the product of NeoPrecis-Immuno-I and NeoPrecis-Immuno-II, thereby integrating both MHC-I and MHC-II pathways. Unless otherwise specified, NeoPrecis-Immuno-dual was used as the default metric in our experiments of ICI response prediction.

We explored three methods to aggregate these mutation-centric scores into a comprehensive tumor-centric immunogenicity score. The NeoPrecis-LandscapeSum method involved summing the NeoPrecis-Immuno scores across all identified neoantigens, providing a straightforward cumulative measure. The NeoPrecis-LandscapeCCF approach leveraged the cancer cell fraction (CCF) by applying a weighted sum based on mutation prevalence as inferred from PyClone analysis, thus incorporating the tumor's cellular composition into the immunogenicity assessment. The NeoPrecis-LandscapeClone method integrated tumor clonality to offer a more nuanced perspective. Using PyClone, mutations were grouped into clusters according to their prevalence in the tumor and mutation size. Within each cluster, NeoPrecis-Immuno scores were summed separately to derive cluster-specific immunogenicity for MHC-I and MHC-II, which were then integrated using multiplication (summation and maximum were also evaluated for comparison). The MHC-integrated, cluster-centric immunogenicity scores were then averaged across clusters, weighted by their prevalence, to calculate the tumor-centric immunogenicity score (Fig. 6a). The three NeoPrecis-Landscape scores were log-transformed to facilitate interpretation within a more manageable range of values.

## Clonality-aware approaches for benchmarking

We benchmarked NeoPrecis-Landscape against two other approaches that derive information from tumor clonality, the Cauchy-Schwarz index of neoantigens (CSiN)[68] and the immunoediting-based optimized tumor neoantigen load (ioTNL)[69].

The CSiN score is calculated by aggregating the normalized product of variant allele frequency (VAF) and neoantigen load across various quality cutoffs for predicted peptide-MHC (pMHC) binding.

For each mutation, we considered all possible 8 to 11-mer (MHC-I) and 15-mer (MHC-II) mutant peptides paired with their respective MHC alleles. The neoantigen load ($L_i$) for a given mutation $i$ represents the number of these pMHC pairs that surpass a specific MHC-binding prediction rank percentile cutoff ($c$). Following the original CSiN methodology, we employed rank percentile cutoffs of 0.375, 0.5, 0.625, 0.75, 1.25, 1.75, and 2. Mutations with a VAF ($V_i$) below 0.05 were excluded. For samples with over 500 mutations, only the 500 mutations with the highest VAF were retained for analysis. The CSiN score is then computed using the following equation:

$$\text{CSiN} = \frac{\sum_{c \in \{c_0, c_1, .., c_k\}} \log\left(\frac{\sum_{i=1}^{n} I(q(i) < c) \cdot \frac{V_i}{V_c} \times \frac{L_i}{L_c}}{\sum_{i=1}^{n} I(q(i) < c)}\right)}{k} \quad (26)$$

where $k$ is the total number of cutoffs used, $n$ is the number of mutations considered for the patient, and $q(i)$ is the best rank percentile among all pMHCs for mutation $i$.

The ioTNL score is calculated by summing the product of cancer cell fraction (CCF) and neoantigen load for subclones identified through PyClone analysis[67] that exhibit evidence of immune elimination. An immune elimination score ($L_j/S_j$) is calculated for each subclone to quantify its potential for immune elimination; only subclones with a score below a defined cutoff ($e$) are included. The ioTNL score is computed using the following equation:

$$\text{ioTNL} = \sum_{j=1}^{m} I\left(\frac{L_j}{S_j} < e\right) \cdot L_j \times \text{CCF}_j \quad (27)$$

where $m$ is the total number of subclones, $L_j$ is the neoantigen load in subclone $j$, $S_j$ is the number of nonsynonymous mutations in subclone $j$, and $e$ is the predefined cutoff for the immune elimination score. The selection of the cutoff ($e = 1.4$) was empirically determined to maximize the predictive performance of ioTNL across our datasets. Although this data-driven approach carries a risk of overfitting and potential information leakage, no established biological rationale was available to guide this selection.

### Survival analysis

For survival analysis, patients were stratified into two groups based on their target metric scores, with one group comprising the top half and the other comprising the bottom half of the distribution. Due to differences in data availability, we used overall survival (OS) for melanoma and progression-free survival (PFS) for NSCLC. Kaplan-Meier survival curves were generated, and log-rank tests were performed to assess statistical differences between the two groups.

### Tumor heterogeneity assessment

To evaluate tumor heterogeneity, we calculated the Gini index at two levels: cluster mutation size and cluster prevalence, capturing diversity in both dimensions. The Gini index quantifies inequality, with a value of 0 indicating perfect equality (all clusters have equal mutation counts or prevalence) and a value of 1 indicating maximum inequality (a single cluster contains all mutations or has a prevalence of 1). The Gini index ($G$) is computed as follows, where $n$ represents the number of clusters and $x$ denotes either mutation size or prevalence:

$$G = \frac{\sum_{i=1}^{n} \sum_{j=1}^{n} |x_i - x_j|}{2n \cdot \sum_{i=1}^{n} x_i} \quad (28)$$

To classify tumors into homogeneous and heterogeneous groups, we applied a two-dimensional approach based on prevalence Gini index (pGini) and size Gini index (sGini). For each cancer type, tumors with pGini ≤ median and sGini ≤ median were categorized as heterogeneous, while tumors with pGini > median and sGini > median were classified as homogeneous.

### Batch effect evaluation

To assess potential batch effects across the combined immunotherapy cohorts, we performed two complementary analyses: distribution-based comparisons and mutation profile analysis.

**TMB distribution analysis.** For each cancer type, we categorized patients by response status (responder and non-responder). Within each cancer type and response category, we performed pairwise Mann-Whitney U tests comparing TMB distributions across cohorts. P-values were adjusted using the Benjamini-Hochberg false discovery rate (FDR) correction. Cohort pairs with FDR-adjusted p-values < 0.05 were considered to have significantly different TMB distributions.

**Mutation profile analysis.** We constructed a binary mutation matrix (presence/absence) for all protein-coding genes across all patients. Principal component analysis (PCA) was performed separately for melanoma and NSCLC. To quantify cohort-specific clustering, we calculated silhouette scores for the top 10 principal components using cohort labels as the clustering variable. Silhouette scores range from −1 to 1, where values near 0 indicate no meaningful clustering structure, and negative values suggest that cohort assignment does not reflect natural groupings in the data, both supporting the absence of cohort-driven clustering.

**Performance evaluation across cohorts.** To assess whether batch effects impacted model performance, we conducted within-cohort validation by calculating AUROC for each score separately for each cohort. We then performed 5-fold cross-validation incorporating cohort as a categorical covariate in a logistic regression framework.

### Reporting summary

Further information on research design is available in the Nature Portfolio Reporting Summary linked to this article.

## Data availability

The processed data generated in this study are provided in the Supplementary Data files, including cross-reactive peptide triplets, CEDAR immunogenicity data, processed NCI gastrointestinal cancer cohort data, and the ICI cohort metadata with analysis results. Publicly available datasets analyzed in this study were obtained from the Immune Epitope Database (IEDB, https://www.iedb.org), VDJdb (https://vdjdb.cdr3.net), and the Cancer Epitope Database and Analysis Resource (CEDAR, https://cedar.iedb.org). The mutation-centric immunogenicity data (NCI dataset) were obtained from the supplementary materials of Parkhurst et al. (https://doi.org/10.1158/2159-8290.CD-18-1494). Clinical and genomic data from the reanalyzed ICI cohorts were obtained with the following accession numbers: Hugo et al. (SRP090294 [https://trace.ncbi.nlm.nih.gov/Traces/study/?acc=SRP090294], SRP067938), Van Allen et al. (SRP011540 [https://trace.ncbi.nlm.nih.gov/Traces/study/?acc=SRP011540]), Snyder et al. (SRP072934 [https://trace.ncbi.nlm.nih.gov/Traces/study/?acc=SRP072934]), Riaz et al. (SRP094781 [https://trace.ncbi.nlm.nih.gov/Traces/study/?acc=SRP094781]), Liu et al. (SRP011540 [https://trace.ncbi.nlm.nih.gov/Traces/study/?acc=SRP011540]), Ravi et al. (SRP413932 [https://trace.ncbi.nlm.nih.gov/Traces/study/?acc=SRP413932]), Anagnostou et al. (SRP238904 [https://trace.ncbi.nlm.nih.gov/Traces/study/?acc=SRP238904]), Rizvi et al. (SRP064805 [https://trace.ncbi.nlm.nih.gov/Traces/study/?acc=SRP064805]).

## Code availability

Codes for conducting immunogenicity prediction and neoantigen landscape evaluation are deposited at both GitHub (https://github.

com/cartercompbio/NeoPrecis) and Zenodo (https://doi.org/10.5281/zenodo.17959604).

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

## Acknowledgements

This work was funded by the Mark Foundation Emerging Leader Award #18-022-ELA, NCI grant R01CA269919, and support from NCI grant U24CA248138 to H. Carter. Computational resources were supported by an infrastructure grant 2P41GM103504-11. The ICI cohorts were collected from several published studies. The melanoma cohort from Hugo

and colleagues was obtained from the Sequence Read Archive (SRA) under accessions SRP090294 and SRP067938. The melanoma cohorts from Van Allen et al. and Liu et al. were obtained from dbGaP under accession phs000452, supported by the National Human Genome Research Institute (NHGRI) Large Scale Sequencing Program, Grant U54 HG003067 to the Broad Institute (PI, Lander). The melanoma cohort from Riaz and colleagues was obtained from SRA under accession SRP094781. The melanoma cohort from Snyder and colleagues was obtained from dbGaP under accession phs001041; we thank Martin Miller at Memorial Sloan Kettering Cancer Center (MSKCC) for his assistance with the NetMHC server, Agnes Viale and Kety Huberman at the MSKCC Genomics Core, Annamalai Selvakumar and Alice Yeh at the MSKCC HLA typing laboratory for their technical assistance, and John Khoury for assistance in chart review. The NSCLC cohort from Rizvi and colleagues was obtained from dbGaP under accession phs000980. We thank the members of the Thoracic Oncology Service and the Chan and Wolchok labs at MSKCC for helpful discussions. We thank the Immune Monitoring Core at MSKCC, including L. Caro, R. Ramsawak, and Z. Mu, for exceptional support with processing and banking peripheral blood lymphocytes. We thank P. Worrell and E. Brzostowski for help in identifying tumor specimens for analysis. We thank A. Viale for superb technical assistance. We thank D. Philips, M. van Buuren, and M. Toebes for help performing the combinatorial coding screens. The data presented in this paper are tabulated in the main paper and in the supplementary materials. This work was supported by the Geoffrey Beene Cancer Research Center (MDH, NAR, TAC, JDW, AS), the Society for Memorial Sloan Kettering Cancer Center (MDH), Lung Cancer Research Foundation (WL), Frederick Adler Chair Fund (TAC), The One Ball Matt Memorial Golf Tournament (EBG), Queen Wilhelmina Cancer Research Award (TNS), The STARR Foundation (TAC, JDW), the Ludwig Trust (JDW), and a Stand Up To Cancer-Cancer Research Institute Cancer Immunology Translational Cancer Research Grant (JDW, TNS, TAC). Stand Up To Cancer is a program of the Entertainment Industry Foundation administered by the American Association for Cancer Research. The NSCLC cohort from Anagnostou and colleagues was obtained from dbGaP under accession phs001940, supported in part by US National Institutes of Health grant CA121113. The NSCLC cohort from Ravi and colleagues was obtained from dbGaP under accession phs002822. We express our deep gratitude to the patients and families whose participation enabled this study. We further thank the respective sequencing centers at Yale University, Johns Hopkins University, and the Broad Institute of MIT and Harvard for processing the whole exome and RNA-seq data presented here. Funding for this study was provided by a Stand Up To Cancer - American Cancer Society Lung Cancer Dream Team Translational Research Grant (Grant Number: SU2C-AACR-DT17-15). Stand Up to Cancer is a program of the Entertainment Industry Foundation. Research grants are administered by the American Association for Cancer Research, the scientific partner of SU2C. This work was additionally supported by The Mark Foundation for Cancer Research (Grant Number: 19-029-MIA) Expanding Therapeutic Options for Lung Cancer (EXTOL) project.

## Author contributions

Conceptualization: K.H.L., T.J.S., M.Z., H.C. Methodology: K.H.L. Investigation: K.H.L., T.J.S. Visualization: K.H.L. Funding acquisition: H.C. Project administration: H.C. Supervision: M.Z., H.C. Writing – original draft: K.H.L. Writing – review & editing: T.J.S., M.Z., H.C.

## Competing interests

MZ is an advisor to the board of Invectys Inc. All other authors declare no competing interests.
