## [Transparent Peer Review file · Nature Communications]

NeoPrecis: Enhancing Immunotherapy Response Prediction through Integration of Qualified Immunogenicity and Clonality-Aware Neoantigen Landscapes

Corresponding Author: Professor Hannah Carter

Version 0:

Reviewer comments:

Reviewer #1

(Remarks to the Author)

Many papers have tried to use HLA-presentation predictors as a proxy for neoantigen identifications for the field of immunotherapy. However, few has explicitly trained for missense mutation recognitions by the immune systems. This paper filled the gap by proposing a machine-learning method and a tailored-training dataset in identifying neoantigens arising from missense-mutations. This also provided a more nuanced method in patient stratification in immunotherapy. My recommendation at this stage is revision.

Major comments

1. Multiple immunotherapy cohorts (5 melanoma cohorts and 3 NSCLC cohorts) were combined to do analyses. How did you adjust for cohort-to-cohort differences? Or sequence batch effects?
2. The training set that was created was created in mind of finding differences in T cells' ability to differentiate peptides with only 1 amino acid difference. How many unique T cells were included per pMHC triplet? Are these T cells public clones or private clones of patients? My concern about this method is whether this can be generalized or is it only accounting for specific T cells that are private a few people?
3. Training data size for HLA class II is small. How many alleles were you able to train for? How did you account for overfit? Do you see any skew towards specific alleles for either HLA class I or II?

Minor comments:

1. Please rearrange locations of figures 3B, 3C, 3D to make them read from left to right and top to bottom for easier readability

(Remarks on code availability)

Reviewer #2

(Remarks to the Author)

The authors describe a computational framework that predicts the immunogenicity of neoantigens, with the option to include tumor clonality as an additional feature to enhance prediction of response to ICI therapy. The authors show on-par performance of the immunogenicity model when compared with previously published studies using similar approaches. However, the improvement in performance is not strikingly different and could be highly dependent on the test dataset. I believe the novelty of the tool could be more focused on the incorporation of clonality information, and how this additional information can be leveraged to improve predictions across more tumor cohorts. Please see detailed comments below:

Major comments:

1. The authors highlight the use of allele-specific MHC binding motifs as well as position-weighted pooling as novel features for their immunogenicity prediction model. How were the MHC binding motifs and positional weights derived or calculated? Given that motifs are generally derived from peptides that bind or are presented by the respective allele, how does the model handle alleles with no training data and thus no clear binding motif?

2.Distance based on BLOSUM62 was used throughout the manuscript for multiple comparisons (e.g., comparing to the geometric distance calculated by NeoPrecis-Immuno). Could the authors show similar comparisons using PMBEC, or further explain why PMBEC was not considered a more biologically relevant distance matrix?

3.For NeoPrecis-Immuno's evaluation on the CEDAR dataset (internal test set), the positive-to-negative ratio is roughly 1:1 (line 165). This ratio seems quite different from other benchmark datasets (e.g., test sets used in ICERFIRE and PRIME). Does this ratio reflect the true proportion of peptides that can be recognized by T cells? Additionally, for the external test set, the authors mention aggregation methods across multiple peptides for determining mutation site immunogenicity. Have the authors tried and compared multiple aggregation strategies, and are performance results dependent on the aggregation method? Do the authors mask positive peptides found in training prior to aggregation and evaluation? Could the authors provide and comment on model performance comparisons at an allele+length-specific level?

4.Given the relatively small dataset sizes, could the authors clarify the number of alleles and peptide lengths present in the training/testing data? How reliable is the model when generalizing to alleles not seen during training (do such cases exist in the test set)?

5.When calculating a tumor-centric immunogenicity score using NeoPrecis-Immuno predictions (line 298), the authors directly multiply MHC-I and MHC-II predictions for the same mutation. This seems to prioritize a single mutation that activates both a class I and class II response, rather than two separate mutations where one activates class I and the other activates class II. Can the authors elaborate on why they believe this is the appropriate strategy?

6.The authors primarily focused on demonstrating the incorporation of tumor clonality information for generating a tumor-level overall score. However, it would be interesting to see whether incorporating clonality at the mutation level (without using DNA VAF as a proxy) could help improve neoantigen immunogenicity predictions.

Minor comments:

- When describing the construction of the triplet dataset, it was a bit confusing that the authors describe the positives as non-immunogenic mutated peptides and the negatives as immunogenic (lines 145–147), while in Figure S1A, the seed and positive peptides are described as having corresponding TCRs. Given that negative peptides were randomly selected, it is unclear why they are described as immunogenic mutated peptides.

- It would be appreciated if the authors labeled the training dataset sizes for the models (e.g., how many seed peptides were used from IEDB/VDJdb, and how many data points in CEDAR were retained for training, given that it was also used for internal validation).

- For evaluation metrics, could the authors also report PPVn in the context of neoantigen immunogenicity prioritization?

- How are the core positions for Class II peptides determined?

- In Figure 3D, could the authors comment on the outlier and why motif-embedding did not improve performance in this case?

- In Figure 4C/S5B, there doesn't appear to be a significant gain when comparing A+R with A+P+R. Can the authors comment on why this might be the case? Also, the AUROC values for A+R and A+P+R are extremely close — it would be helpful to have the values labeled directly.

- In Figure 4D, could the authors also show the feature importance of binding/presentation predictions from NetMHCpan?

(Remarks on code availability)

The code is well documented with examples and usage.

Version 1:

Reviewer comments:

Reviewer #1

(Remarks to the Author)

The authors have answered my questions. I have no more questions for the authors.

(Remarks on code availability)

Reviewer #2

(Remarks to the Author)

I appreciate the authors' thorough responses to my previous comments. I have no further remarks.

(Remarks on code availability)

REVIEWER COMMENTS

Reviewer #1 (Remarks to the Author):

Many papers have tried to use HLA-presentation predictors as a proxy for neoantigen identifications for the field of immunotherapy. However, few has explicitly trained for missense mutation recognitions by the immune systems. This paper filled the gap by proposing a machine-learning method and a tailored-training dataset in identifying neoantigens arising from missense-mutations. This also provided a more nuanced method in patient stratification in immunotherapy. My recommendation at this stage is revision.

We thank the reviewer for the helpful comments that have strengthened the manuscript. We have conducted more analyses to address the concern about model generalizability and robustness, including cohort-to-cohort batch correction in ICI response prediction, CDR3 coverage in the training set, and model generalizability across unobserved MHC alleles.

Major comments:

1. Multiple immunotherapy cohorts (5 melanoma cohorts and 3 NSCLC cohorts) were combined to do analyses. How did you adjust for cohort-to-cohort differences? Or sequence batch effects?

Thank you for raising this important concern. Although we applied a uniform bioinformatics protocol for somatic mutation calling and HLA typing from raw sequences, we acknowledge that sequencing-related batch effects between cohorts could potentially influence our findings. To address this, we systematically evaluated **potential batch effects across cohorts** and assessed **their impact on immunotherapy response prediction performance**.

Batch effect:

Since our study focuses on somatic mutations and immunotherapy response, we evaluated batch effects at two levels: **tumor mutation burden (TMB)** and **mutation profiles**. For TMB, we performed pairwise Mann-Whitney U tests with FDR correction to compare TMB distributions across cohorts within the same cancer type and response category. While most cohort pairs showed no significant differences, we identified several exceptions, including Riaz vs. Liu and Riaz vs. Van Allen in melanoma non-responders, as well as Ravi vs. Rizvi in NSCLC non-responders (**Fig. R1A**). For mutation profiles, we constructed a binary presence/absence mutation matrix for all coding genes across patients and performed principal component analysis (PCA) separately for each cancer type. Though there are some outliers, the top two principal components revealed no obvious cohort-specific clustering, with low explained variance (**Fig. R1B**). Furthermore, silhouette scores for the top 10 PCs were -0.36 for melanoma and -0.30 for NSCLC, indicating minimal cohort-driven structure in the mutation data.

Conclusion on Batch effect. Together these analyses suggest that **batch effects related to nonsynonymous somatic mutations are modest across cohorts**.

Batch correction:

To further evaluate whether cohort-related batch effects impact model performance, we conducted two complementary analyses. First, we assessed **performance within individual cohorts** and found that NP-LandscapeClone outperformed TMB in 4 out of 5 melanoma cohorts, while NP-LandscapeSum outperformed TMB in the Ravi NSCLC cohort (**Fig. R1C**). Second, we incorporated cohort as a covariate in a 5-fold cross-validation framework to directly adjust for potential batch effects. The results remained consistent with our original findings (**Fig. S11a**), with NP-LandscapeClone achieving the best performance in melanoma and NP-LandscapeSum in NSCLC (**Fig. R1D**).

In summary, our batch effect analysis reveals only mild cohort-to-cohort variation, and when accounting for this variation, the result remained unchanged with NeoPrecis consistently outperforming other methods. We have added these batch effect analyses to the supplementary materials to provide greater transparency regarding data heterogeneity across cohorts.

Manuscript updates:

Updated Figures: Fig. R1A-B has been added as Fig. S7a-b.; Fig. R1C-D has been added as Fig. S11b-c.

Updated Results:

Line 315-320

To assess cohort compatibility, we evaluated TMB distributions and mutation profiles (**Methods**). Pairwise Mann-Whitney U tests with FDR correction identified significant TMB differences in only a few cohort pairs (**Fig. S7a**). Principal component analysis of binary mutation profiles revealed no obvious cohort-specific clustering, with silhouette scores of -0.36 for melanoma and -0.30 for NSCLC indicating minimal cohort-driven structure (**Fig. S7b**).

Line 393-398

To ensure this superior performance was not driven by cohort-specific effects, we assessed models within individual cohorts. NeoPrecis-LandscapeClone outperformed TMB in 4 of 5 melanoma cohorts, while NeoPrecis-LandscapeSum outperformed TMB in the Ravi NSCLC cohort (**Fig. S11b**). When incorporating cohort as a covariate in cross-validation, the relative performance of all approaches remained unchanged (**Fig. S11c**), confirming the superiority of NeoPrecis across cohorts.

Updated Methods: Line 883-901

To assess potential batch effects across the combined immunotherapy cohorts, we performed two complementary analyses: distribution-based comparisons and mutation profile analysis.

TMB distribution analysis: For each cancer type, we categorized patients by response status (responder and non-responder). Within each cancer type and response category, we performed pairwise Mann-Whitney U tests comparing TMB distributions across cohorts. P-values were adjusted using the Benjamini-Hochberg false discovery rate (FDR) correction. Cohort pairs with FDR-adjusted p-values < 0.05 were considered to have significantly different TMB distributions.

Mutation profile analysis: We constructed a binary mutation matrix (presence/absence) for all protein-coding genes across all patients. Principal component analysis (PCA) was performed separately for melanoma and NSCLC. To quantify cohort-specific clustering, we calculated silhouette scores for the top 10 principal components using cohort labels as the clustering variable. Silhouette scores range from -1 to 1, where values near 0 indicate no meaningful clustering structure, and negative values suggest that cohort

assignment does not reflect natural groupings in the data, both supporting the absence of cohort-driven clustering.

Performance evaluation across cohorts: To assess whether batch effects impacted model performance, we conducted within-cohort validation by calculating AUROC for each score separately for each cohort. We then performed 5-fold cross-validation incorporating cohort as a categorical covariate in a logistic regression framework.

Fig. R1. Batch effect evaluation across ICI cohorts.

(A) Pairwise Mann-Whitney U test comparing TMB distributions across cohorts within the same cancer type and response category for melanoma (left) and NSCLC (right). Heatmaps show FDR-corrected p-values. (new Fig. S7a)

(B) Distribution of top 2 principal components of binary mutation profiles for NSCLC (top) and melanoma (bottom). Each point represents a patient, colored by cohort. (new Fig. S7b)

(C) Cohort-specific AUROC performance comparing TMB, CSiN, ioTNL, NP-LandscapeSum, NP-LandscapeCCF and NP-LandscapeClone within individual cohorts. (new Fig. S11b)

(D) AUROC performance using 5-fold cross-validation repeated 100 times with cohort incorporated as a covariate for melanoma (left) and NSCLC (right). (new Fig. S11c)

2. The training set that was created was created in mind of finding differences in T cells' ability to differentiate peptides with only 1 amino acid difference. How many unique T cells were included per pMHC triplet? Are these T cells public clones or private clones of patients? My concern about this method is whether this can be generalized or is it only accounting for specific T cells that are private a few people?

This is an important point regarding model generalizability. To predict similarity in terms of T-cell recognition between wild-type and mutant peptides, we identified cross-reactive peptide pairs from pMHC-TCR binding datasets. In our study, cross-reactive peptides are defined as peptides presented by the same MHC allele and recognized by the same TCR CDR3 sequence. This definition captures peptides that are truly cross-reactive and solely due to the change in amino acid composition. Under this strict criterion, we curated a dataset of 1,153 MHC-I and 261 MHC-II cross-reactive peptide pairs.

Although our model does not explicitly incorporate CDR3 sequences, analyzing CDR3 coverage provides important insights into generalizability. We calculated **the number of distinct CDR3 sequences that bind to each cross-reactive peptide pair** and found that approximately 50% of pairs are recognized by multiple CDR3 sequences (**Fig. R2A**). Since our model focuses on single amino acid changes, we further examined **CDR3 support for peptide pairs sharing the same amino acid substitution**. We found that more than 50% of these substitutions are supported by multiple independent CDR3 sequences (**Fig. R2B**). This multi-CDR3 support suggests that cross-reactivity is driven by generalizable peptide features rather than specific CDR3 sequences. Our model captures these intrinsic peptide properties by learning from diverse TCR-peptide binding patterns. Furthermore, the model performed well at predicting T-cell reactivity in the NCI dataset which is not constrained to the subset of pMHCs or CDR3s on which NeoPrecis was trained and fine-tuned. In our response to the next comment, we further address the generalizability across unobserved HLA alleles.

We acknowledge that due to the vast diversity of the available TCR repertoire ($2.5 \times 10^7 - 3.8 \times 10^8$), our dataset captures only a limited fraction of possible CDR3 sequences. We have added this analysis to the **Results** section and note that expanding the dataset with additional pMHC-TCR binding data would further strengthen the model's generalizability.

Manuscript updates:

Updated Figures: Fig. R2 has been added as Fig. S3a

Updated Results: Line 190–196

To assess model generalizability, we evaluated TCR diversity in the training data and performance on unobserved MHC alleles. For TCR diversity, we analyzed CDR3 coverage in the cross-reactive peptide triplet dataset and found that approximately 50% of cross-reactive peptide pairs are recognized by multiple distinct CDR3 sequences, with more than 50% of amino acid substitutions supported by multiple independent CDR3s (**Fig. S3a**). This multi-CDR3 support indicates that the training data captures generalizable cross-reactivity patterns rather than isolated clone-specific interactions.

Fig. R2. Cumulative distribution of the number of distinct CDR3 sequences supporting (A) each cross-reactive peptide pair and (B) each amino acid substitution in the triplet dataset. (new Fig. S3a)

3. Training data size for HLA class II is small. How many alleles were you able to train for? How did you account for overfit? Do you see any skew towards specific alleles for either HLA class I or II?

We appreciate this important question about model generalizability across HLA alleles. Due to the strict criteria for the triplet dataset (derived from TCR-pMHC binding data), allele coverage is indeed limited, with only 15 MHC-I and 7 MHC-II alleles represented. However, the fine-tuning dataset (immunogenicity data from CEDAR) provides broader coverage with 88 MHC-I and 23 MHC-II alleles. To address the challenge of limited allele diversity, we represent MHC alleles using pMHC binding motifs rather than allele identity, which enables the model to capture a wider range of MHC alleles and generalize to unobserved alleles by capturing shared peptide-binding preferences.

We evaluated model generalizability across MHC alleles in two complementary ways: performance on high-frequency versus low-frequency alleles, and performance on unobserved alleles. Both the triplet and CEDAR datasets exhibit allele frequency skew reflecting population-level biases (Fig. R3A-B). For instance, HLA-A*02:01, which is prevalent in European/Caucasian populations, is similarly dominant in our datasets. To assess whether this skew leads to performance bias, we compared model performance on the top 5 most frequent alleles (high-frequency group) versus all other alleles (low-frequency group) in the CEDAR test set. This analysis was restricted to MHC-I due to sample size limitations for MHC-II. NeoPrecis-Immuno, PRIME, and ICERFIRE outperformed NetMHCpan and showed consistent performance across both groups, whereas DeepNeo exhibited substantially worse performance in the low-frequency allele group (Fig. R3C). These results demonstrate that our motif-based allele representation effectively mitigates potential overfitting to over-represented alleles.

To directly evaluate generalizability to unobserved alleles, we stratified mutations in the NCI dataset by the number of unobserved alleles—those absent from training data. The CEDAR test set contained only 2 unobserved alleles each for MHC-I and MHC-II (Fig. R4A), necessitating the use of NCI for this evaluation. In NCI, 28% of mutations had unobserved MHC-I alleles, while nearly all mutations had at least one unobserved MHC-II allele (Fig. R4B). Although performance decreased in the unobserved allele group, NeoPrecis-Immuno outperformed other predictors for MHC-I and performed comparably for MHC-II (Fig. R4C). We note that alleles unobserved in our training data may have been observed in other predictors' training sets, potentially contributing to their performance in these cases. This result shows that our model generalizes effectively across unobserved MHC alleles.

Manuscript updates:

Updated Figures: Fig. R4 has been added as Fig. S3b-d

Updated Results: Line 196–206

For MHC allele generalization, we stratified mutations in the NCI dataset by the number of unobserved alleles—those absent from training data. The CEDAR test set contained only 2 unobserved alleles each for MHC-I and MHC-II (**Fig. S3b**), necessitating the use of NCI for this evaluation. In NCI, 28% of mutations had unobserved MHC-I alleles, while nearly all mutations had at least one unobserved MHC-II allele (**Fig. S3c**). Although performance decreased in the unobserved allele group, NeoPrecis-Immuno outperformed other predictors for MHC-I and performed comparably for MHC-II (**Fig. S3d**). We note that alleles unobserved in our training data may have been observed in other predictors' training sets, potentially contributing to their performance in these cases. These results demonstrate that NeoPrecis-Immuno generalizes effectively across unobserved MHC alleles and that the training data represents diverse TCR recognition patterns, though limited training diversity remains a constraint.

Fig. R3. Performance of high-frequency versus low-frequency alleles.

(A) Proportion distribution of the top 10 frequent alleles in the triplet dataset.

(B) Proportion distribution of the top 10 frequent alleles in the CEDAR training set.

(C) Performance (AUROC and AUPRC) of immunogenicity predictors stratified by allele frequency. High-frequency alleles are defined as the top 5 alleles; low-frequency alleles comprise all others. Performance metrics were calculated using bootstrapping (1,000 iterations).

Fig. R4. Performance evaluation of unobserved alleles. (new Fig. S3b-d)

- (A) Venn diagram showing allele overlap between the cross-reactive triplet dataset, CEDAR training set, and CEDAR testing set for MHC-I (top) and MHC-II (bottom).
- (B) Cumulative distribution of the number of unobserved alleles per mutation in the NCI dataset for MHC-I (left) and MHC-II (right). Unobserved alleles are those absent from the NeoPrecis-Immuno training data.
- (C) Performance (AUROC) of immunogenicity predictors stratified by unobserved allele groups for MHC-I (left) and MHC-II (right). Mutations were grouped by the number of unobserved alleles (0 vs. > 0 for MHC-I; ≤ 5 vs. > 5 for MHC-II). AUROC distributions were calculated using bootstrapping (1,000 iterations). In the boxplots, the center line represents the median, boxes indicate the interquartile range (IQR; 25th–75th percentile), and whiskers extend to the most extreme data points within 1.5× the IQR. Outliers are depicted as individual circles.

Minor comments:

1. Please rearrange locations of figures 3B, 3C, 3D to make them read from left to right and top to bottom for easier readability

We thank the reviewer for this suggestion. We have rearranged the layout of **Fig. 3b-d** to follow a left-to-right, top-to-bottom reading order, which improves the clarity and flow of the figure.

Reviewer #2 (Remarks to the Author):

The authors describe a computational framework that predicts the immunogenicity of neoantigens, with the option to include tumor clonality as an additional feature to enhance prediction of response to ICI therapy. The authors show on-par performance of the immunogenicity model when compared with previously published studies using similar approaches. However, the improvement in performance is not strikingly different and could be highly dependent on the test dataset. I believe the novelty of the tool could be more focused on the incorporation of clonality information, and how this additional information can be leveraged to improve predictions across more tumor cohorts. Please see detailed comments below:

We thank this reviewer for the helpful suggestions which have improved the clarity of our work. We have addressed all concerns about model generalizability, including MHC allele and peptide length bias, through comprehensive analyses using unobserved allele and length stratification. Additionally, we have emphasized the novel contribution of clonality integration by demonstrating the coordination of MHC-I and MHC-II pathways in tumor subclonal immunity, with significant performance improvement when using multiplicative integration of MHC-I and MHC-II predictions at the subclonal level.

Major comments:

1. The authors highlight the use of allele-specific MHC binding motifs as well as position-weighted pooling as novel features for their immunogenicity prediction model. How were the MHC binding motifs and positional weights derived or calculated? Given that motifs are generally derived from peptides that bind or are presented by the respective allele, how does the model handle alleles with no training data and thus no clear binding motif?

We thank the reviewer for this important question regarding MHC binding motifs and position weights. We acknowledge that our original description of NeoPrecis-Immuno lacked sufficient detail, and we have substantially revised the **Methods** section to provide a comprehensive explanation of how these components are derived and implemented.

Calculation of MHC binding motifs and positional weights:

Input preparation: pMHC complexes are first predicted using an MHC-binding predictor, and the 9-mer core binding region is identified as the input sequence. For each neoantigen, we obtain paired wild-type (WT) and mutant (MT) peptide sequences represented as one-hot encoded matrices: $p^{WT}, p^{MT} \in \mathbb{R}^{L \times 20}$, where $L=9$ is the core binding region length and each row is a one-hot vector over 20 amino acids.

MHC binding motif representation: To represent MHC alleles in a generalizable manner, we constructed binding motifs from NetMHCpan predictions. We randomly sampled 100,000 peptides from the human peptidome (9-mers for MHC-I, 15-mers for MHC-II) and predicted binding affinity for each available allele. Peptides passing the binding rank threshold (2 for MHC-I, 10 for MHC-II) were retained, and their 9-mer binding cores were used to compute position-specific amino acid frequencies. The resulting binding motif $M \in \mathbb{R}^{L \times 20}$ serves as the MHC allele representation, where each position contains the frequency distribution over 20 amino acids.

Amino acid embedding: One-hot encoded amino acids are first transformed using the BLOSUM62 substitution matrix, then projected into a latent space via a single-layer neural network:

$$E^{WT} = f_{embed}(P^{WT}) \in \mathbb{R}^{L \times d_a}$$

$$E^{MT} = f_{embed}(P^{MT}) \in \mathbb{R}^{L \times d_a}$$

$$E^{motif} = f_{embed}(M) \in \mathbb{R}^{L \times d_a}$$

where $d_a=2$ is the amino acid embedding dimension.

Motif enrichment: To capture the complex interactions between peptide sequences and MHC alleles, we introduce a motif-dependent convolutional layer. The model learns $K=8$ convolutional kernels $W_k^{conv} \in \mathbb{R}^{d_a \times d_a}$, which are modulated by the MHC motif embedding:

$$C_k = E^{motif} W_k^{conv} \in \mathbb{R}^{L \times d_a}, k = 1, \dots, K$$

These motif-enriched kernels are then applied position-wise to peptide embeddings:

$$H_k^{WT} = C_k \odot E^{WT} \in \mathbb{R}^L$$

$$H_k^{MT} = C_k \odot E^{MT} \in \mathbb{R}^L$$

where \odot denotes element-wise multiplication followed by summation across the embedding dimension. The K kernel responses are concatenated and projected to the peptide embedding dimension:

$$E_{enriched}^{WT} = W^{proj} [H_1^{WT}, \dots, H_K^{WT}] \in \mathbb{R}^{L \times d_p}$$

$$E_{enriched}^{MT} = W^{proj} [H_1^{MT}, \dots, H_K^{MT}] \in \mathbb{R}^{L \times d_p}$$

where $W^{proj} \in \mathbb{R}^{d_p \times K}$ is a learnable projection matrix and $d_p=2$ is the peptide embedding dimension.

Position-weighted peptide pooling: To aggregate position-enriched embeddings into fixed-length peptide representations, we employ position-weighted pooling. Each of the 9 core binding positions has a learnable weight w_i (separate parameters for MHC-I and MHC-II):

$$v^{WT} = \sum_{i=1}^L w_i E_{enriched,i}^{WT} \in \mathbb{R}^{d_p}$$

$$v^{MT} = \sum_{i=1}^L w_i E_{enriched,i}^{MT} \in \mathbb{R}^{d_p}$$

where $E_{enriched,i}$ denotes the embedding at position i . This design allows the model to emphasize structurally important positions in the MHC binding groove.

Geometric representation: To capture the distinction between wild-type and mutant peptides in a way that reflects TCR cross-reactivity, we construct a geometric representation in the peptide embedding space. The geometric transformation consists of three components:

$$v^{origin} = v^{WT} \in \mathbb{R}^{d_p}$$

$$v^{direction} = \frac{v^{MT} - v^{WT}}{\|v^{MT} - v^{WT}\|_2 + \varepsilon} \in \mathbb{R}^{d_p}$$

$$d = \|v^{MT} - v^{WT}\|_2 \in \mathbb{R}$$

where v^{origin} represents the wild-type peptide position, $v^{direction}$ is the normalized direction of mutation in the embedding space, d is the Euclidean distance between wild-type and mutant embeddings, and ε is a small constant to prevent division by zero.

Cross-reactivity distance (CRD): To account for sequence context, we compute a scaling factor using a two-layer neural network:

$$s = f_{scale}([v^{origin}; v^{direction}])$$

The CRD is then computed as:

$$CRD = d \cdot \sigma(s)$$

where σ is the sigmoid function. Higher CRD values indicate lower predicted TCR cross-reactivity between wild-type and mutant peptides.

Immunogenicity prediction: The final immunogenicity prediction integrates CRD with MHC binding prediction:

$$y_{immuno} = \sigma(W_{out}[CRD; MHC_{binding}] + b_{out})$$

This produces an immunogenicity score between 0 and 1.

Model training: The NeoPrecis-Immuno model was trained in two stages. In the first stage, it was trained on the cross-reactive peptide triplet dataset using contrastive learning with triplet loss to optimize CRD:

$$\mathcal{L}_{triplet} = \max(0, CRD(a, p) - CRD(a, n) + margin)$$

where a represents an anchor peptide, p is a cross-reactive peptide, n is a non-cross-reactive peptide, and the margin was set to 1. In the second stage, the model was fine-tuned on the CEDAR immunogenicity dataset using binary cross-entropy loss:

$$\mathcal{L}_{BCE} = -\frac{1}{N} [y_i \log(\hat{y}_i) + (1 - y_i) \log(1 - \hat{y}_i)]$$

where y_i is the true immunogenicity label and \hat{y}_i is the predicted score. During fine-tuning, all layers were frozen, except for the scaling factor and classifier, allowing the model to focus on adjusting immunogenicity predictions. Training was conducted for 100 epochs with a learning rate of 0.005, using the AdamW optimizer.

Unobserved alleles:

The reviewer also raises a critical consideration regarding alleles absent from the training data. While our motif-based representation is designed to generalize across alleles, evaluating performance on unobserved alleles is essential to validate this approach.

Atkins et al. demonstrated that NetMHCpan maintains strong performance on alleles not included in its training set, which supports the reliability of binding motifs derived from NetMHCpan predictions. Building on this foundation, we directly evaluated NeoPrecis-Immuno performance on unobserved alleles. The CEDAR test set contained only 2 unobserved alleles each for MHC-I and MHC-II (**Fig. R4A**), necessitating the use of the NCI dataset for this evaluation.

Since the NCI dataset is a mutation-level dataset where each mutation can potentially bind to any of an individual's MHC alleles, we calculated the proportion of unobserved alleles per mutation. We found that most MHC-I alleles were represented in the training data, with only 28% of mutations involving unobserved MHC-I alleles, whereas nearly all mutations had at least one unobserved MHC-II allele (**Fig. R4B**). We then stratified mutations by the number of unobserved alleles and compared model performance. Although performance decreased in the unobserved allele group, NeoPrecis-Immuno outperformed other predictors for MHC-I and performed comparably for MHC-II (**Fig. R4C**). We note that alleles unobserved in our training data may have been observed in other predictors' training sets, potentially contributing to their performance in these cases. These results demonstrate that our motif-based representation successfully generalizes to unobserved alleles despite limited training data for these alleles.

Manuscript updates:

Updated Figures: Fig. R4 has been added as Fig. S3b-d

Updated Results: Line 196–206

For MHC allele generalization, we stratified mutations in the NCI dataset by the number of unobserved alleles—those absent from training data. The CEDAR test set contained only 2 unobserved alleles each for MHC-I and MHC-II (**Fig. S3b**), necessitating the use of NCI for this evaluation. In NCI, 28% of mutations had unobserved MHC-I alleles, while nearly all mutations had at least one unobserved MHC-II allele (**Fig. S3c**). Although performance decreased in the unobserved allele group, NeoPrecis-Immuno outperformed other predictors for MHC-I and performed comparably for MHC-II (**Fig. S3d**). We note that alleles unobserved in our training data may have been observed in other predictors' training sets, potentially contributing to their performance in these cases. These results demonstrate that NeoPrecis-Immuno generalizes effectively across unobserved MHC alleles and that the training data represents diverse TCR recognition patterns, though limited training diversity remains a constraint.

Updated Methods: The details of NeoPrecis model architecture have been add to Line 570–646

Reference:

Atkins TK, Solanki A, Vasmatzis G, Cornette J, Riedel M. Evaluating NetMHCpan performance on non-European HLA alleles not present in training data. *Front Immunol.* 2024 Jan 16;14:1288105. doi: 10.3389/fimmu.2023.1288105. PMID: 38292493; PMCID: PMC10825027.

Fig. R4. Performance evaluation of unobserved alleles. (new Fig. S3b-d)

- (A) Venn diagram showing allele overlap between the cross-reactive triplet dataset, CEDAR training set, and CEDAR testing set for MHC-I (top) and MHC-II (bottom).
- (B) Cumulative distribution of the number of unobserved alleles per mutation in the NCI dataset for MHC-I (left) and MHC-II (right). Unobserved alleles are those absent from the NeoPrecis-Immuno training data.
- (C) Performance (AUROC) of immunogenicity predictors stratified by unobserved allele groups for MHC-I (left) and MHC-II (right). Mutations were grouped by the number of unobserved alleles (0 vs. > 0 for MHC-I; ≤ 5 vs. > 5 for MHC-II). AUROC distributions were calculated using bootstrapping (1,000 iterations). In the boxplots, the center line represents the median, boxes indicate the interquartile range (IQR; 25th–75th percentile), and whiskers extend to the most extreme data points within 1.5× the IQR. Outliers are depicted as individual circles.

2. Distance based on BLOSUM62 was used throughout the manuscript for multiple comparisons (e.g., comparing to the geometric distance calculated by NeoPrecis-Immuno). Could the authors show similar comparisons using PMBEC, or further explain why PMBEC was not considered a more biologically relevant distance matrix?

We thank the reviewer for this insightful suggestion to compare our model with PMBEC, an amino acid substitution matrix specifically derived from peptide-MHC binding data. We have now incorporated PMBEC comparisons into our analysis.

Comparison of amino acid representations:

We obtained the PMBEC encoding from *Kim et al.* and compared three amino acid representation methods: BLOSUM62, PMBEC, and NeoPrecis-Immuno embeddings. For BLOSUM62 and PMBEC, we applied PCA to extract the first two principal components as two-dimensional embeddings for visualization (**Fig. R5A**). The three representation methods yield distinctly different embedding structures, reflecting their different derivation approaches and biological contexts. For example, in NeoPrecis-Immuno, histidine (H) was positioned closer to tyrosine (Y) and phenylalanine (F), likely due to their shared aromatic rings, while tryptophan (W) was positioned further away, possibly reflecting its larger size. These differences between NeoPrecis and PMBEC suggests distinct amino acid interaction patterns in TCR recognition versus MHC presentation.

Performance comparison on immunogenicity prediction:

We evaluated the predictive power of amino acid distances derived from each method for immunogenicity prediction, following the approach in Fig. 3a (**Fig. R5B**). We tested two distance calculation strategies: (1) Euclidean distance in PCA-reduced space (top 2 PCs) to obtain continuous distance distributions (BLOSUMDist, PMBECDist), and (2) direct substitution scores from the matrices (BLOSUMDist(sub), PMBECDist(sub)). The substitution distance derived from NeoPrecis-Immuno (SubDist) outperformed all baseline metrics regardless of distance calculation method.

Manuscript updates:

In the main manuscript, we report only the PC-derived distance metrics (BLOSUMDist, PMBECDist; **Fig. 3a**) to maintain continuous distance distributions, which is particularly important for **Fig. S2a**. We have added PMBEC as a baseline comparison and updated the **Methods** section to clarify the amino acid distance calculation approach.

Updated Figures: Fig. 3a (component analysis), Fig. S2a (distribution of PMBEC distance), Fig. S4a (Correlation between components)

Updated Results: Line 211–212

BLOSUM62 (BLOSUMDist) and PMBEC (PMBECDist) distances were included as baselines for comparison.

Updated Methods: Line 680–686

For baseline comparison, we used BLOSUM62 and PMBEC substitution matrices. BLOSUM62 is an amino acid substitution matrix derived from aligned protein sequences, while PMBEC is specifically derived from peptide-MHC binding data. Since substitution matrices provide discrete scores for amino acid pairs, we applied PCA to each matrix to extract the first two principal components as two-dimensional amino acid

embeddings, enabling continuous distance calculations. We then computed distances as the Euclidean distance between wild-type and mutated amino acids in this reduced space (BLOSUM62Dist, PMBECDist).

Reference:

Kim Y, Sidney J, Pinilla C, Sette A, Peters B. Derivation of an amino acid similarity matrix for peptide: MHC binding and its application as a Bayesian prior. *BMC Bioinformatics*. 2009 Nov 30;10:394. doi: 10.1186/1471-2105-10-394. PMID: 19948066; PMCID: PMC2790471.

Fig. R5. Comparison of amino acid embeddings and distance metrics.

(A) Two-dimensional amino acid embeddings derived from BLOSUM62 (left), PMBEC (center), and NeoPrecis (right). For BLOSUM62 and PMBEC, the top 2 principal components from PCA are shown. NeoPrecis embeddings are learned during model training.

(B) AUROC comparison of amino acid distance metrics and incrementally constructed NeoPrecis-Immuno components for MHC-I (left) and MHC-II (right) predictions on the NCI dataset. Distance metrics include: BLOSUM_Dist (Euclidean distance in top 2 PC space), BLOSUM_Dist(sub) (negative substitution score), PMBEC_Dist (Euclidean distance in top 2 PC space), PMBEC_Dist(sub) (negative substitution score), SubDist (NeoPrecis substitution distance), SubPosDist (SubDist with position weighting), GeoDist (SubPosDist with motif enrichment), and CRD (GeoDist with sigmoid scaling). (new Fig. 3a)

3. For NeoPrecis-Immuno's evaluation on the CEDAR dataset (internal test set), the positive-to-negative ratio is roughly 1:1 (line 165). This ratio seems quite different from other benchmark datasets (e.g., test sets used in ICERFIRE and PRIME). Does this ratio reflect the true proportion of peptides that can be recognized by T cells? Additionally, for the external test set, the authors mention aggregation methods across multiple peptides for determining mutation site immunogenicity. Have the authors tried and compared multiple aggregation strategies, and are performance results dependent on the aggregation method? Do the authors mask positive peptides found in training prior to aggregation and evaluation? Could the authors provide and comment on model performance comparisons at an allele+length-specific level?

We appreciate these important concerns about real-world applicability of neoantigen prediction models. We have conducted additional analyses and revised the manuscript to address the positive-to-negative ratio, aggregation strategies, and allele- and length-specific performance.

Positive-to-negative ratio:

In this study, we aimed to predict T-cell recognition of MHC-presented peptides. However, the true positive rate is difficult to estimate and varies considerably across studies due to differences in experimental methods and selection criteria.

The experimental workflow significantly impacts observed positive rates. Studies following a "top-down" approach—first selecting MHC-binding peptides, then testing for T-cell responses—typically observe positive rates of 20-40%. In the CEDAR dataset, the overall positive rate using multimer/tetramer assays is 22.2% for MHC-I (493/2,223), while Alban et al. reported 39.0% (196/502) in non-small cell lung cancer using a similar top-down strategy. In contrast, "bottom-up" approaches that perform T-cell assays without MHC binding pre-selection yield much lower rates. The NCI dataset exemplifies this: ELISPOT screening without MHC filtering yielded only 0.6% positive (41/7,384), increasing to 3.3% (36/1,088) when restricted to predicted MHC binders. Our model follows the biologically sound top-down principle by integrating MHC binding with T-cell cross-reactivity metrics.

We acknowledge that the 50% positive rate in our CEDAR test set does not reflect realistic clinical scenarios, resulting from filtering for predicted MHC binders and selecting neoantigens absent from other predictors' training sets. For fair comparison with existing immunogenicity predictors, we examined the test set compositions used in their publications: PRIME used 25.6% positive (1,282/4,958) and ICERFIRE used 20.8% positive (631/3,033). To match these benchmarks, we rebalanced samples through bootstrapping to adjust our positive rate to 25%. The resulting AUROC and AUPRC rankings remained consistent across predictors (**Fig. R6A**), demonstrating that our model's performance is robust to class imbalance.

Aggregation strategies:

To predict mutation-level immunogenicity from allele-specific scores, we evaluated multiple aggregation strategies that account for individual MHC genotypes. Each strategy involves two key decisions: whether to apply allele masking and which aggregation method to use.

Allele masking: Masking excludes alleles predicted not to bind the target peptide, based on NetMHCpan %rank thresholds (≤ 2 for MHC-I, ≤ 10 for MHC-II). Let A_{all} denote all MHC alleles for an individual and $A_{bind} \subseteq A_{all}$ denote binding alleles. Without masking, all alleles contribute to aggregation; with masking, only A_{bind} is considered.

Aggregation methods: For allele-specific scores s_a , we evaluated five aggregation methods.

- Mean: $s_{mut} = \frac{1}{|A|} \sum_{a \in A} s_a$
- Harmonic mean: $s_{mut} = \frac{|A|}{\sum_{a \in A} (s_a + \epsilon)^{-1}}$
- Weighted average: $s_{mut} = \frac{\sum_{a \in A} w_a s_a}{\sum_{a \in A} w_a}$, where w_a is binding affinity from NetMHCpan
- Best binding: $s_{mut} = s_{a^*}$, where $a^* = \operatorname{argmin}_{a \in A} \operatorname{rank}(a)$
- Maximum: $s_{mut} = \max_{a \in A} s_a$

We compared all ten combinations (with/without masking \times five aggregation methods) on the NCI dataset and found that masked maximum approach performed most consistently across both MHC classes and different immunogenicity metrics (**Fig. R6B**; **Fig. S5b**).

Allele- and length-specific performance

Our model generalizes across alleles and peptide lengths by leveraging NetMHCpan predictions. We represent alleles using binding motifs and standardize inputs to 9-mer core binding regions regardless of original peptide length. However, to assess potential biases across different alleles and lengths, we conducted peptide-level performance comparisons on the CEDAR test set.

Due to limited sample sizes, we analyzed alleles and lengths separately. We identified alleles and lengths with at least 40 peptides, then used bootstrapping to compute performance metrics across models. DeepNeo was excluded from this analysis as it only accepts 9-mer inputs.

Compared to the NetMHCpan baseline, NeoPrecis-Immuno demonstrated the most consistent performance, outperforming the baseline in 4 of 5 alleles and both lengths (**Fig. R6C**). In contrast, PRIME and ICERFIRE showed substantial variation across alleles, with PRIME outperforming the baseline in only 2 of 5 alleles and ICERFIRE in only 1 allele. These results indicate that NeoPrecis-Immuno exhibits less allele-specific bias and maintains robust performance across diverse MHC contexts.

Manuscript updates:

Updated Figures: Fig. S2d (performance of rebalanced testing set), Fig. S5b (aggregation analysis)

Updated Results: Line 177–180

We also rebalanced samples through bootstrapping to achieve a 1:3 positive-to-negative ratio, matching the test set balance used in PRIME and ICERFIRE benchmarks and better reflecting realistic immunogenicity rates, which yielded similar performance rankings (**Fig. S2d**).

Updated Methods: The above aggregation method has been updated to Line 748–764

Reference:

Alban TJ, Riaz N, Parthasarathy P, Makarov V, Kendall S, Yoo SK, Shah R, Weinhold N, Srivastava R, Ma X, Krishna C, Mok JY, van Esch WJE, Garon E, Akerley W, Creelan B, Aanur N, Chowell D, Geese WJ, Rizvi NA, Chan TA. Neoantigen immunogenicity landscapes and evolution of tumor ecosystems during immunotherapy with nivolumab. *Nat Med.* 2024 Nov;30(11):3209-3222. doi: 10.1038/s41591-024-03240-y. Epub 2024 Sep 30. PMID: 39349627; PMCID: PMC12066197.

Fig. R6. Evaluation of NeoPrecis-Immuno under class rebalance, aggregation approach, and allele or length stratification.

- (A) Performance (AUROC and AUPRC) for each predictor on rebalanced MHC-I samples ($n = 438$; positive-to-negative ratio adjusted to 1:3 through bootstrapping). Performance distributions were calculated using bootstrapping (1,000 iterations). (**new Fig. S2d**)
- (B) The heatmap displays the negative log p-values from the association tests (Mann–Whitney U test) between each approach and T-cell activity across various TCR recognition metrics. Annotations indicate the rank within each MHC class and for each metric. (**new Fig. S5b**)
- (C) AUROC of each predictor stratified by allele (left) or peptide length (right) for MHC-I samples with ≥ 40 data points per group. Performance was evaluated using bootstrapping (1,000 iterations).

4. Given the relatively small dataset sizes, could the authors clarify the number of alleles and peptide lengths present in the training/testing data? How reliable is the model when generalizing to alleles not seen during training (do such cases exist in the test set)?

We thank the reviewer for this important question about dataset composition and model generalizability to unobserved alleles.

Dataset composition:

Due to strict criteria for identifying cross-reactive peptide pairs in the triplet training dataset, allele coverage is indeed limited, with only 15 MHC-I and 7 MHC-II alleles represented. However, the fine-tuning dataset (immunogenicity data from CEDAR) provides substantially broader coverage with 88 MHC-I and 23 MHC-II alleles (**Fig. R4A in response to Reviewer 2 Comment 1**).

For the peptide length, NeoPrecis uses the binding core region derived from NetMHCpan as input, where the binding core length is consistently 9-mer regardless of the original peptide length. To assess potential length bias, we examined the distribution of original peptide lengths in our datasets. In the CEDAR training set, 9-mer peptides predominate for MHC-I alleles, while 15-mer peptides are most common for MHC-II alleles (**Fig. R7**). Since DeepNeo only accepts 9-mer peptides, we focused on 9-mer performance in the main figures (**Fig. 2c**). For a more comprehensive length evaluation, we analyzed both 9-mer and 10-mer peptides in the CEDAR test set as described in our response to the previous comment (**Fig. R6C in response to Reviewer 2 Comment 3**). Comparison with the NetMHCpan baseline revealed no obvious performance bias across these peptide lengths.

Evaluation on unobserved alleles:

The CEDAR test set contains only 2 unobserved alleles each for MHC-I and MHC-II (**Fig. R4A in response to Reviewer 2 Comment 1**), limiting our ability to assess generalization within this dataset. To more comprehensively evaluate performance on unobserved alleles, we conducted additional analysis using the independent NCI dataset, as described in our **response to Comment 1**.

We stratified mutations by the number of unobserved alleles and compared model performance across groups. We found that most MHC-I alleles were represented in the training data, with only 28% of mutations involving unobserved MHC-I alleles, whereas nearly all mutations had at least one unobserved MHC-II allele (**Fig. R4B in response to Reviewer 2 Comment 1**). Although performance decreased in the unobserved allele group, NeoPrecis-Immuno outperformed other predictors for MHC-I and performed comparably for MHC-II (**Fig. R4C in response to Reviewer 2 Comment 1**). We note that alleles unobserved in our training data may have been observed in other predictors' training sets, potentially contributing to their performance in these cases. This result shows that our model generalizes effectively across unobserved MHC alleles.

Manuscript updates:

Updated Figures: Fig. R4 has been added as Fig. S3b-d

Updated Results: Line 196–206

For MHC allele generalization, we stratified mutations in the NCI dataset by the number of unobserved alleles—those absent from training data. The CEDAR test set contained only 2 unobserved alleles each for MHC-I and MHC-II (**Fig. S3b**), necessitating the use of NCI for this evaluation. In NCI, 28% of mutations

had unobserved MHC-I alleles, while nearly all mutations had at least one unobserved MHC-II allele (**Fig. S3c**). Although performance decreased in the unobserved allele group, NeoPrecis-Immuno outperformed other predictors for MHC-I and performed comparably for MHC-II (**Fig. S3d**). We note that alleles unobserved in our training data may have been observed in other predictors' training sets, potentially contributing to their performance in these cases. These results demonstrate that NeoPrecis-Immuno generalizes effectively across unobserved MHC alleles and that the training data represents diverse TCR recognition patterns, though limited training diversity remains a constraint.

Fig. R7. Distribution of original peptide lengths in the CEDAR training set.

5. When calculating a tumor-centric immunogenicity score using NeoPrecis-Immuno predictions (line 298), the authors directly multiply MHC-I and MHC-II predictions for the same mutation. This seems to prioritize a single mutation that activates both a class I and class II response, rather than two separate mutations where one activates class I and the other activates class II. Can the authors elaborate on why they believe this is the appropriate strategy?

We thank the reviewer for this thoughtful question about our strategy for integrating MHC-I and MHC-II immunogenicity predictions. We evaluate the MHC integration at both the mutation level and the subclone level.

Mutation level:

Our multiplication approach prioritizes mutations that activate both CD4+ (MHC-II) and CD8+ (MHC-I) T-cell responses. Evidence demonstrates that CD8+ T cells require CD4+ T-cell help at many stages of differentiation and maintenance through T-T cooperation based on associative recognition of antigens. In further support of what our group believes to be a tenet of protective T cell immunity, a 2024 paper by Espinosa-Carrasco *et al.* reported that effective anti-tumor immunity requires a functional triad of CD4+ T cells, CD8+ T cells, and dendritic cells. Critically, this coordinated response depends on the presentation of both CD4 and CD8 neoantigens within the same antigen presenting cell. Under this model, a single mutation capable of activating both MHC-I and MHC-II responses is more likely to elicit coordinated immunity across the tumor cell population. The multiplication strategy reflects this biological principle by assigning higher scores to mutations with dual immunogenic potential.

To determine the optimal integration strategy, we compared three aggregation methods: multiplication ($I \times II$), summation ($I + II$), and maximum ($\max(I, II)$). All methods show comparable performance in AUROC and AUPRC (**Fig. R8A-B**) at the mutation level.

Subclone level:

We evaluated the same three MHC-I and MHC-II integration strategies at the subclone level. Clonal analysis was performed using PyClone, which grouped mutations into clusters representing tumor subclones. Within each cluster, NeoPrecis-Immuno scores were summed separately to derive cluster-specific immunogenicity for MHC-I and MHC-II, which were then integrated using multiplication, summation, or maximum. The MHC-integrated cluster immunogenicity scores were then averaged across clusters, weighted by their prevalence, to calculate the tumor-centric immunogenicity score.

Multiplication consistently outperformed other methods in both melanoma and NSCLC (**Fig. R8C-D**). This superior performance at the subclone level provides additional biological support for the coordination between CD4+ T cell, CD8+ T cell, and dendritic cell. Since tumor cells within the same subclone share mutations, dendritic cells in that subclone can present multiple neoantigens on both MHC-I and MHC-II, enabling coordinated dual-class T-cell responses against the same tumor subclone.

Conclusion:

Given that multiplication outperformed other approaches at subclone levels, and considering the biological principle that coordinated CD4+ and CD8+ responses enhance anti-tumor immunity, we selected multiplication as our integration strategy. This approach prioritizes mutations capable of activating both MHC pathways, which may be particularly important in tumors where such coordinated responses are critical for therapeutic efficacy.

Manuscript updates:

Updated Figures: Fig. R8A-B has been added as Fig. S8b; Fig. R8C-D has been added as Fig. S10.

Updated Results:

Line 340–358:

To conduct a more comprehensive mutation-level evaluation, we calculated a tumor-centric immunogenicity score, NeoPrecis-LandscapeSum, by summing NeoPrecis-Immuno predictions across all mutations (**Methods**). For each mutation, the MHC-dual NeoPrecis-Immuno score was derived by multiplying the MHC-I and MHC-II NeoPrecis-Immuno predictions. This multiplication strategy prioritizes mutations capable of activating both CD4+ and CD8+ T-cell responses. Effective anti-tumor immunity requires coordinated CD4+ T-cell, CD8+ T-cell, and dendritic cell responses, which depends on presenting both MHC-I and MHC-II neoantigens within the same antigen-presenting cell⁴³. Given intratumoral heterogeneity, a single mutation activating both pathways is more likely to elicit coordinated immunity than relying on separate mutations that may not co-occur in the same cell. We compared multiplication ($I \times II$) with summation ($I + II$) and maximum ($\max(I, II)$) aggregation methods. While all methods showed comparable performance at the mutation level (**Fig. S8b**), multiplication was selected based on its superior performance at the subclone level (see below, **Fig. S10**). This integrated approach demonstrated superior performance over TMB and TNB in both melanoma (AUROC = 0.66) and NSCLC (AUROC = 0.75) (**Fig. 5d**). Given the better performance of MHC-dual multiplication, we refer to the tumor-centric immunogenicity score as MHC-dual multiplication in subsequent analyses. In survival analyses (**Methods**), patients in the top half of the tumor-centric score showed significantly better outcomes than those in the bottom half (**Fig. 5e**). NeoPrecis-LandscapeSum outperformed TMB in melanoma and showed comparable performance in NSCLC (**Fig. S8c**).

Line 378–385:

We evaluated the same three MHC-I and MHC-II integration strategies (multiplication, summation, maximum) at the subclone level, where subclone-level MHC-I and MHC-II scores were calculated separately then integrated (**Methods**). Multiplication consistently outperformed other methods in both melanoma and NSCLC (**Fig. S10**). This superior performance at the subclone level provides additional biological support for the coordination between CD4+ T cell, CD8+ T cell, and dendritic cell. Since tumor cells within the same subclone share mutations, dendritic cells in that subclone can present multiple neoantigens on both MHC-I and MHC-II, enabling coordinated dual-class T-cell responses against the same tumor subclone.

Updated Discussion: Line 479-483

These findings align with our results, showing that combining MHC-I and MHC-II predictions improves ICI response prediction at the subclone level (coordinated immune responses against tumor subclones presenting multiple neoantigens on both MHC classes; **Fig. S10**) and shows comparable performance at the mutation level (single mutations activating dual pathways; **Fig. 5b, Fig. S8b**).

Updated Methods: Line 828-834

The NeoPrecis-LandscapeClone method integrated tumor clonality to offer a more nuanced perspective. Using PyClone, mutations were grouped into clusters according to their prevalence in the tumor and mutation size. Within each cluster, NeoPrecis-Immuno scores were summed separately to derive cluster-specific immunogenicity for MHC-I and MHC-II, which were then integrated using multiplication (summation and maximum were also evaluated for comparison). The MHC-integrated, cluster-centric immunogenicity scores were then averaged across clusters, weighted by their prevalence, to calculate the tumor-centric immunogenicity score (**Fig. 6a**).

Reference:

Espinosa-Carrasco G, Chiu E, Scrivo A, Zumbo P, Dave A, Betel D, Kang SW, Jang HJ, Hellmann MD, Burt BM, Lee HS, Schietinger A. Intratumoral immune triads are required for immunotherapy-mediated elimination of solid tumors. *Cancer Cell*. 2024 Jul 8;42(7):1202-1216.e8. doi: 10.1016/j.ccell.2024.05.025. Epub 2024 Jun 20. PMID: 38906155; PMCID: PMC11413804.

Fig. R8. Performance evaluation of mutation-level and subclone-level combination of MHC-I and MHC-II immunogenicity.

- (A) AUROC comparison of NeoPrecis-LandscapeSum scores (mutation level) using MHC-I-only, MHC-II-only, or MHC-dual (combined MHC-I and MHC-II). For MHC-dual, three aggregation methods were compared: multiplication ($I \times II$), summation ($I + II$), and maximum ($\max(I, II)$). (new Fig. S8b)
- (B) AUPRC comparison of NeoPrecis-LandscapeSum scores (mutation level). (new Fig. S8b)
- (C) AUROC comparison of NeoPrecis-LandscapeClone scores (subclone level). (new Fig. S10)
- (D) AUPRC comparison of NeoPrecis-LandscapeClone scores (subclone level). (new Fig. S10)

6. The authors primarily focused on demonstrating the incorporation of tumor clonality information for generating a tumor-level overall score. However, it would be interesting to see whether incorporating clonality at the mutation level (without using DNA VAF as a proxy) could help improve neoantigen immunogenicity predictions.

We thank the reviewer for this excellent suggestion to explore mutation-level clonality. We obtained cancer cell fraction (CCF) data from Muller *et al.*, for the NCI dataset and conducted additional analyses to assess whether CCF improves mutation-level immunogenicity prediction compared to DNA VAF.

We first examined the correlation between CCF and DNA VAF. The Pearson correlation was only 0.236 (Fig. R9A), suggesting that DNA VAF may not be an adequate proxy for mutation clonality, likely due to confounding factors such as copy number alterations and tumor purity.

We evaluated CCF in two contexts. First, we replaced DNA VAF with CCF in the NeoPrecis-Integrated model—a logistic regression model incorporating clonality, RNA VAF, RNA expression, PHBR-II, and NeoPrecis-Immuno scores as features. Despite the low correlation between DNA VAF and CCF, cross-validation performance was comparable between the two models (Fig. R9B), suggesting that clonality information contributes similarly regardless of whether it is measured by CCF or DNA VAF.

Second, we tested multiplicative integration by combining NeoPrecis-Immuno scores with either CCF or DNA VAF, analogous to our tumor-level approach. Neither CCF nor DNA VAF integration improved upon NeoPrecis-Immuno alone (**Fig. R9C**). These findings indicate that mutation-level clonality provides limited additional benefit for immunogenicity prediction when high-quality immunogenicity scores are already available, in contrast to tumor-level analyses where clonality effectively prioritizes clonal neoantigens.

Fig. R9. Incorporating clonality at the mutation level.

- (A) Correlation between DNA AF and CCF in the NCI dataset. r is Pearson correlation coefficient.
- (B) Comparison of integrated models (A+P+R) using either DNA VAF or CCF as the DNA abundance metric. Performance was evaluated using 4-fold cross-validation repeated 100 times.
- (C) Performance (AUROC and AUPRC) of multiplicative integration combining NeoPrecis-Immuno scores with either CCF or DNA AF.

Minor comments:

1. When describing the construction of the triplet dataset, it was a bit confusing that the authors describe the positives as non-immunogenic mutated peptides and the negatives as immunogenic (lines 145–147), while in Figure S1A, the seed and positive peptides are described as having corresponding TCRs. Given that negative peptides were randomly selected, it is unclear why they are described as immunogenic mutated peptides.

We appreciate the opportunity to clarify this source of confusion. The underlying principle of our pre-training approach is to model the distance between wild-type and mutant peptides in terms of T-cell recognition. We leverage cross-reactivity in peptide-TCR interactions to represent peptide similarity: cross-reactive peptides (those recognized by the same TCR) are considered similar, while peptides not sharing TCR recognition are considered dissimilar. This learned distance metric is then applied to immunogenicity prediction, where mutant peptides similar to their wild-type counterparts (high cross-reactivity) are predicted to be less immunogenic, while dissimilar mutant peptides (low cross-reactivity) are predicted to be more immunogenic.

In the original manuscript, we used "positive" to refer to peptides that bind the same TCR as the seed peptide, which indeed represents close distance and lower predicted immunogenicity. We recognize this terminology was confusing, as "positive" could be misinterpreted as "immunogenic."

We have revised the terminology throughout the manuscript for clarity:

"TCR-binding triplet" → "Cross-reactive peptide triplet"

"Positive peptide" → "Cross-reactive peptide" (shares TCR binding with seed)

"Negative peptide" → "Non-cross-reactive peptide" (does not share TCR binding)

Under this revised terminology, **Figure S1a** now clearly shows that the seed and cross-reactive peptides share corresponding TCRs, while non-cross-reactive peptides are randomly selected from peptides that do not bind these TCRs. This framework ensures that the model learns to identify peptides likely to elicit distinct T-cell responses (non-cross-reactive, thus potentially immunogenic) versus those likely to cross-react (similar to self, thus less immunogenic).

Manuscript updates:

Updated Figures: Fig. S1a

Updated Results: Line 143–159

We implemented a two-stage training process using TCR-pMHC binding data from IEDB⁴⁸ and VDJdb⁴⁹ (**Supplementary Data 1**) and T-cell assay data from CEDAR⁵⁰ (**Supplementary Data 2**). In the first stage, we constructed a cross-reactive peptide triplet dataset from binding data, where each triplet consists of a seed peptide, a cross-reactive peptide (peptide sharing TCR binding with the seed), and a non-cross-reactive peptide (peptide not sharing TCR binding). We curated 1,153 MHC-I and 261 MHC-II cross-reactive pairs and sampled 10 non-cross-reactive peptides for each pair, yielding 11,530 MHC-I and 2,610 MHC-II triplets for training. The model was trained to minimize the distance between cross-reactive pairs (seed and cross-reactive peptides) and maximize the distance between non-cross-reactive pairs (seed and non-cross-reactive peptides) (**Fig. S1a, Methods**). In the context of a tumor, mutant peptides more similar to wild-type (cross-reactive with the same TCRs) are less likely to be immunogenic because T cells recognizing these self-similar epitopes are typically eliminated during central tolerance. In contrast, more dissimilar mutants (less cross-reactive) are more likely to be immunogenic, as T cells capable of recognizing these neoantigens escape thymic negative selection and remain available in the peripheral repertoire. Notably, the geometric distance calculated by NeoPrecis-Immuno outperformed the BLOSUM62 and PMBEC (an amino acid similarity matrix for peptide-MHC binding)⁵¹ distances in differentiating cross-reactive and non-cross-reactive peptide pairs (**Fig. 2b, Fig. S2a**).

Updated Methods: Line 521–537

To train the immunogenicity model to distinguish mutated peptides from wild-type peptides, we constructed a dataset of cross-reactive peptide triplets. Each triplet consists of a seed peptide (a reference peptide), a cross-reactive peptide (sharing TCR binding with the seed), and a non-cross-reactive peptide (not sharing TCR binding) (**Fig. S1a**). The model was trained to minimize the distance between seed-cross-reactive pairs (similar peptides) while maximizing the distance between seed-non-cross-reactive pairs (dissimilar peptides). To mimic the effect of single amino acid substitutions, the difference between both cross-reactive and non-cross-reactive pairs was constrained to one Hamming distance. Cross-reactive pairs were sampled from each peptide-MHC complex group to ensure functional similarity between the seed and cross-reactive peptides. All possible pairs were sampled, yielding 1,153 MHC-I and 261 MHC-II cross-reactive pairs.

For each cross-reactive pair, non-cross-reactive peptides were randomly selected to be one Hamming distance away from the seed but absent from the cross-reactive set, which comprised peptides within the same peptide-MHC complex group. A BLOSUM62 matrix was used as a prior, ensuring that the substitution between seed and non-cross-reactive peptides resulted in a negative BLOSUM62 score, indicating substantial sequence dissimilarity. To enhance dataset diversity, we applied 10-fold generation, ultimately producing 11,530 MHC-I and 2,610 MHC-II triplets (**Supplementary Data 1**).

2. It would be appreciated if the authors labeled the training dataset sizes for the models (e.g., how many seed peptides were used from IEDB/VDJdb, and how many data points in CEDAR were retained for training, given that it was also used for internal validation).

We thank the reviewer for this request. This information was previously provided in the **Methods** section, and we have now added key sample sizes to the **Results** section as well for improved clarity and accessibility.

Cross-reactive peptide triplet dataset (pre-training):

The dataset comprises 1,153 MHC-I and 261 MHC-II seed peptides. For each seed peptide, we identified one cross-reactive peptide (sharing the same TCR) and sampled 10 non-cross-reactive peptides. In total, the pre-training dataset contains 11,530 MHC-I triplets and 2,610 MHC-II triplets.

CEDAR immunogenicity dataset (fine-tuning):

After filtering for predicted MHC binding (netMHCpan %rank ≤ 2 for MHC-I and ≤ 10 for MHC-II), the dataset comprised 4,176 MHC-I and 87 MHC-II samples. These were split into training (75%), validation (10%), and testing (15%) sets, yielding 3,143 training, 415 validation, and 618 testing samples for MHC-I, and 54 training, 11 validation, and 22 testing samples for MHC-II. From the initial testing set of 640 samples, we retained 438 unique peptides that were confirmed to be absent from the training data of all comparison methods (NeoPrecis-Immuno, PRIME, DeepNeo, and ICERFIRE) for unbiased evaluation.

Manuscript updates:

Updated Results:

Line 147–149:

We curated 1,153 MHC-I and 261 MHC-II cross-reactive pairs and sampled 10 non-cross-reactive peptides for each pair, yielding 11,530 MHC-I and 2,610 MHC-II triplets for training.

Line 160–165:

In the second stage, we fine-tuned the model using T-cell assay data with T-cell activation labels, enabling it to learn the immunogenicity of target peptides (**Methods**). We curated 4,176 MHC-I and 87 MHC-II samples predicted to bind MHC (NetMHCpan %rank ≤ 2 for MHC-I and ≤ 10 for MHC-II) from CEDAR and split them into training (75%), validation (10%), and testing (15%) sets. The resulting immunogenicity model was subsequently validated on both an internal and an external testing set to benchmark its performance.

3. For evaluation metrics, could the authors also report PPV_n in the context of neoantigen immunogenicity prioritization?

Thank you for the recommendation. We add PPV@*k* for the neoantigen immunogenicity prioritization, especially for the NCI dataset. We select *k* as the number of true positives.

Manuscript updates:

Updated Figures: Fig. S2e (top), Fig. S6b (bottom)

Updated Results: Line 182–185

For MHC-I, NeoPrecis-Immuno outperformed all other predictors across AUROC, AUPRC, and PPV metrics. For MHC-II, NeoPrecis-Immuno achieved the highest AUROC while performing comparably to DeepNeo in AUPRC and PPV (Fig. 2d, Fig. S2e).

4. How are the core positions for Class II peptides determined?

Core binding regions for both MHC-I and MHC-II peptides are determined using NetMHCpan predictions. In addition to binding affinity and rank scores, NetMHCpan outputs the predicted 9-mer core binding region for each peptide-MHC interaction. This 9-mer core is used as input to our model for both MHC class I and class II peptides, ensuring consistent representation across alleles and peptide lengths. For peptides shorter than 9 residues, the sequence is padded to obtain the 9-mer binding core as specified by NetMHCpan.

5. In Figure 3D, could the authors comment on the outlier and why motif-embedding did not improve performance in this case?

We thank the reviewer for highlighting this outlier, which corresponds to allele A*68:02 at mutant position 3. This case involves only one positive and one negative sample, detailed in the table below.

Label	WT_aa	MT_aa	SubDist	GeoDist	CRD	%rank	immunogenicity
Negative	S	I	2.90	2.79	2.79	1.2	0.39
Positive	N	I	5.56	1.47	1.47	0.13	0.48

In the substitution distance (SubDist), the N to I substitution (positive sample) has a larger distance (5.56) than the S to I substitution (negative sample, 2.90). However, after motif enrichment, the geometric distance (GeoDist) reverses this ranking. This occurs because the motif enrichment applies different scaling factors to embedding dimensions. For A*68:02 position 3, the learned scaling factors are 4.55 for dimension 1 and 2.96 for dimension 2 (**Fig. 3c**). In the embedding space, S and I differ primarily along dimension 1, while N and I differ more along dimension 2 (**Fig. 3b**). Since dimension 1 has a larger scaling factor (4.55 vs. 2.96), the S-to-I distance is amplified more than the N-to-I distance, resulting in a larger GeoDist for the negative sample (2.79 vs. 1.47).

Notably, when MHC binding affinity (%rank) is integrated into the final immunogenicity prediction, the positive sample correctly receives a higher score (0.48 vs. 0.39) despite the lower CRD. This case illustrates that T-cell cross-reactivity distance is not the sole determinant of immunogenicity—here, the strong MHC binding of the positive sample (%rank = 0.13) compensates for its lower CRD, ultimately producing the correct prediction. This example demonstrates the importance of integrating multiple signals, as MHC binding can be the dominant factor in certain cases.

6. In Figure 4C/S5B, there doesn't appear to be a significant gain when comparing A+R with A+P+R. Can the authors comment on why this might be the case? Also, the AUROC values for A+R and A+P+R are extremely close — it would be helpful to have the values labeled directly.

We thank the reviewer for this observation. We have added metric values directly to the bar plots in **Fig. 4c** and **Fig. S6b** (previously S5b) for easier comparison.

Regarding the similar performance of A+R and A+P+R, this occurs because NeoPrecis-Immuno already incorporates MHC binding predictions as a covariate during training and inference, effectively capturing presentation information within the recognition metric. Feature importance analysis supports this interpretation: in the A+P+R model, recognition dominates while presentation contributes minimally; in contrast, when recognition is absent (A+P model), presentation becomes an important predictor (**Fig. R10A-B**). These results demonstrate that integrating multi-dimensional features substantially improves immunogenicity prediction compared to single features alone. NeoPrecis-Immuno serves as a comprehensive metric that efficiently captures both T-cell recognition and MHC presentation information, explaining why explicit inclusion of presentation features provides minimal additional benefit when recognition is already included.

Manuscript updates:

Updated Figures: Fig. 4c (figure below), 4d (Fig. R10A), S6b (in minor comment 3), S6c (Fig. R10B)

Updated Results: Line 289–302

To refine predictions, we integrated the mutation-centric features into a logistic regression model. After performing feature selection, abundance features (A): DNA AF, RNA AF, and RNA expression; presentation feature (P): PHBR; and recognition feature (R): NeoPrecis-Immuno were included in the final integrated multi-dimensional model, NeoPrecis-Integrated (**Fig. S6a, Methods**). Cross-validation repeated 100 times on the NCI dataset demonstrated that the A+P+R and A+R achieved similar performance (average AUROC: 0.848 vs. 0.849 for MHC-I; 0.806 for MHC-II), both substantially outperforming single-feature models (**Fig. 4c, Fig. S6b**). This occurs because NeoPrecis-Immuno already incorporates MHC binding predictions as a covariate, effectively capturing presentation information. Feature importance analysis confirmed that recognition dominates in A+P+R while presentation contributes minimally, whereas presentation becomes important in A+P models (**Fig. 4d, Fig. S6c**). These results demonstrate that integrating multi-dimensional features improves immunogenicity prediction compared to single features alone, with NeoPrecis-Immuno serving as a comprehensive metric that captures both T-cell recognition and presentation information.

Fig. R10. Feature importance in multi-dimensional integrated models. Features were standardized prior to modeling, and importance values were derived from logistic regression coefficients.

(A) Feature importance in the A+P+R integrated model. (new Fig. 4d)

(B) Comparison of feature importance across models using different feature combinations, including A+P, A+R, and A+P+R. (new Fig. S6c)

7. In Figure 4D, could the authors also show the feature importance of binding/presentation predictions from NetMHCpan?

We thank the reviewer for this suggestion. NetMHCpan binding predictions are represented in our analysis through PHBR (harmonic mean of MHC binding ranks across alleles), which aggregates NetMHCpan predictions to the mutation level. PHBR, as the presentation (P) feature, was already included in the original Fig. 4d. We have now also added feature importance comparisons between A+P and A+P+R models in Fig. R10B (new Fig. S6c). As shown, MHC presentation is an important predictor in the A+P model. However, in the A+P+R model, its importance is substantially reduced, as the presentation signal is effectively captured within NeoPrecis-Immuno, which incorporates MHC binding predictions as a covariate.

Manuscript updates:

Updated Figures: Fig. 4d, Fig. S6c (provided in the previous comment)

Updated Results: Line 289–302 (provided in the previous comment)

Reviewer #2 (Remarks on code availability):

The code is well documented with examples and usage.